# Understanding biases in ICESat-2 data due to subsurface scattering using Airborne Topographic Mapper waveform data

Benjamin E. Smith[1], Michael Studinger[2], Tyler Sutterley[1], Zachary Fair[2], and Thomas Neumann[2]

[1]University of Washington Applied Physics Laboratory Polar Science Center, Seattle, WA, 98122, USA
[2]Cryospheric Sciences Laboratory, NASA Goddard Space Flight Center, Greenbelt, MD, 20771, USA

Correspondence to: Benjamin Smith (besmith@uw.edu)

**Abstract.**

The process of laser light reflecting from surfaces made of scattering materials that do not strongly absorb at the wavelength of the laser can involve reflections from hundreds or thousands of individual grains, which can introduce delays in the time between light entering and leaving the surface. These time-of-flight biases depend on the grain size and density of the medium, and so can result in spatially and temporally varying surface height biases estimated from laser altimeters, such as NASA's ICESat-2 (Ice Cloud, and land Elevation Satellite-2) mission. Modelling suggests that ICESat-2 might experience a bias difference as large as 0.1-0.2 m between coarse-grained melting snow and fine-grained wintertime snow (Smith et al., 2018), which exceeds the mission's requirement to measure seasonal height differences to an accuracy better than 0.1 m (Markus et al., 2017). In this study, we investigate these biases using a model of subsurface scattering, laser altimetry measurements form NASA's ATM (Airborne Topographic Mapping) system, and grain-size estimates based on optical imagery of the ice sheet. We demonstrate that distortions in the shapes of waveforms measured using ATM are related to the optical grain size of the surface estimated using optical reflectance measurements, and show that they can be used to estimate an effective grain radius for the surface. Using this effective grain radius as a proxy for the severity of subsurface scattering, we use our model with grain-size estimates from optical imagery to simulate corrections for biases in ICESat-2 data due to subsurface scattering, and demonstrate that, on the basis of large-scale averages, the corrections calculated based on the satellite optical imagery match the biases in the data. This work demonstrates that waveform-based altimetry data can measure the optical properties of granular surfaces, and that corrections based on optical grain-size estimates can correct for subsurface-scattering biases in ICESat-2 data.

## 1 Introduction.

Laser altimetry techniques allow efficient measurement of precise surface elevations for ice sheets and glaciers, both from satellites (Abdalati et al., 2010) and aircraft (MacGregor et al., 2021). Repeated measurements over glaciers and ice sheets

allow the detection of surface elevation changes that show the effects of surface-mass-balance and ice-dynamic processes (Smith et al., 2020), while measurements over floating ice are used to estimate sea ice thickness (Petty et al., 2022) and to infer melt rates beneath ice shelves (Sutterley et al., 2019).  These techniques rely on the altimeter's ability to measure the range to the ice or snow surface with high precision.  Since its launch in late 2018, ICESat-2 has been making high-precision measurements of ice-sheet and glacier elevation.  Unlike the near-infrared (1064-nm) laser used by its predecessor, ICESat,

ICESat-2's laser transmits and receives green light, with a wavelength of 532 nm.  The shorter wavelength allows ICESat-2 to use highly sensitive detectors to measure the arrival time of individual return photons, improving its overall precision and resolution relative to that of ICESat (Brunt et al., 2021; Markus et al., 2017).  At the same time, the choice of a green laser introduces potential biases in its altimetry measurements because ice absorbs green light weakly (Warren and Brandt, 2008), allowing photons to scatter over relatively long distances within the snow before returning to the surface and, potentially, the

satellite. These biases can interfere with ICESat-2's primary mission goals of precisely measuring elevation changes over glaciers, ice sheets, and sea ice (Markus et al., 2017)  because time varying biases in ICESat-2 measurements could produce spurious signals that might be interpreted as ice-sheet elevation changes (Smith et al., 2018).  Likewise, spatially varying biases in ICESat-2 measurements over sea ice might falsely be interpreted as variability in freeboard and thus ice thickness(Harding et al., 2011)(Harding et al., 2011; Smith et al., 2018)

The problem of biases in altimetry data that result from subsurface multiple scattering in snow and ice has been described in previous studies (Harding et al., 2011; Smith et al., 2018).   Light is reflected from snow surfaces primarily by multiple scattering, where each photon scatters off many snow grains before escaping the snowpack (Wiscombe and Warren, 1980; Warren, 1982). When light scatters from granular materials that absorb light strongly, only those photons that have scattered a small number of times escape the surface.  By contrast, light scattering from weakly absorbing granular materials may enter

the surface and scatter from tens or hundreds of grains before escaping again.  The extra distance travelled during these subsurface scattering events delays the return of the photons to the surface, so light escaping the surface includes photons that have travelled a distribution of long and short paths.  A lidar system measuring the range to a weakly absorbing surface will measure returning photons that have a longer mean travel time and a broader distribution of return times than it would from a non-scattering or strongly absorbing surface.  The mean delay of the photons and the shape of the returning pulse (i.e. the

measured waveform in an analog lidar, or the distribution of photon timing in a photon-counting lidar) depend on the scattering properties of the material, with lower densities and coarser grain sizes corresponding to deeper penetration of photons into the snow, broader returns, and longer delay times (Fair et al., 2024).  Light absorption within the scattering medium can also influence time distribution of returning photons, with stronger absorption producing narrower distributions and smaller net delays because photons are more often absorbed by the medium before they can accumulate long delays. The distribution in

time of reflected energy thus can provide information about the optical properties of snow and ice surfaces.

The dependence of return photon timing distribution on ice optical properties has also been explored in recent studies (Smith et al., 2018; Allgaier and Smith, 2021; Hu et al., 2022), including one study where researchers have used predictions from a scattering model to interpret measurements from a hand-carried system to estimate snow and ice optical properties, using a

pulsed laser and a detector pressed against the ice surface, separated by a few centimeters (Allgaier et al., 2022). Although other researchers have noted the potential for these theories to be applied to laser remote-sensing measurements, only a few studies have attempted to infer snow and firn properties based on remotely sensed lidar scattering measurements (Hu et al., 2022; Lu et al., 2022; Harding et al., 2011). More recently, a study using ATM measurements from Northeast Greenland demonstrated an association between apparent elevation differences between green and near-infrared laser-altimetry measurements and grain-size variations (Fair et al., 2024). A second study demonstrated that subsurface scattering of green laser light is associated with negative biases in estimated sea-ice surface elevations, in some cases leading to floating-ice elevations that are apparently below the water surface (Studinger et al., 2024).

In this study, we investigate the scattering properties of Greenland snow and ice surfaces using waveform shapes from an airborne laser altimeter, with the goal of developing a correction for the biases that subsurface scattering can introduce into ICESat-2 data. Although this study is motivated by the need to understand biases in ICESat-2 measurements related to subsurface scattering of green light, data from ICESat-2 are rarely suitable for investigation of subsurface scattering biases, because over rough and sloping surfaces, ICESat-2's 11-m footprint leads to a significant random component in the timing of returned photons, which tends to obscure small changes in the timing distribution associated with subsurface scattering. Slope and roughness tend to be largest in low-elevation regions of Greenland (Nolin and Payne, 2007), which are the same regions where we expect to see the largest subsurface scattering biases. Instead, we use waveform measurements from the ATM airborne laser-altimetry system to test a previously developed model of subsurface scattering (Smith et al., 2018) based on a comparison between the shapes of the returned pulses and pulse shapes expected based on the model. We demonstrate that when we adjust the grain size and surface roughness in the model to match modelled waveforms to measured waveforms we can recover an estimate of the near-surface optical grain size. We test the grain-size estimates from waveform matching by comparing them against grain-size estimates derived from airborne and satellite reflectance measurements. Although this comparison does not suggest a 1:1 linear relationship between waveform-derived grain sizes and reflectance-derived grain sizes, we use ICESat-2 biases calculated for the ATM grain-size estimates as a proxy for direct measurements of ICESat-2 biases to calibrate a correction based on reflectance-derived grain sizes, and demonstrate that the calibrated correction can produce elevation estimates that, averaged over a range of Greenland terrain and surface conditions, are unbiased. Although the results of this study fall short of a correction that could eliminate grain-size-driven biases in ICESat-2 data, we provide a description of some of the advances in satellite remote sensing that would be needed to more adequately address this problem.

## 2. Data

This study is based on waveform data from the ATM lidar systems, grain-size estimates based on the airborne AVIRIS-NG (Airborne Visible/Infrared Imaging Spectrometer, Next-Generation) spectroradiometer, and grain-size estimates based on the spaceborne OLCI (Ocean and Land Colour Instrument) instrument. A summary of measurement locations for the airborne data is presented in section 3.

**2.1 Altimetric waveforms from the Airborne Topographic Mapping lidar systems.**

ATM (the Airborne Topographic Mapping system) makes laser-altimetry measurements using a conically scanning laser that maps elevations beneath an airplane over a swath 40-500 m wide. ATM has made measurements over the Greenland and Antarctic ice sheets since 1993, with an evolving configuration of lasers and measurement strategies that have gradually improved measurement precision and reliability (MacGregor et al., 2021; Krabill et al., 2002). Since 2017, the system has used green (532-nm) lasers with a 1.3-ns pulse duration (full width at half maximum) and a receiver with a bandwidth of around 1 GHz. At a nominal flight elevation of 500 m above ground level the size of the lidar footprint on the surface is ~0.70 m diameter. ATM's configurations include a narrow-swath scanner whose 5º full scan angle makes measurements over a ~40-m swath on the ground at a flight elevation of 500 m, and a wide-swath scanner whose 30º scan angle produces a ~460-m swath.

Many lidars, including both photon-counting instruments such as that used by ICESat-2 and analog instruments such as ATM, can measure the time distribution of light that has reflected off their targets. Photon-counting altimeters measure the distribution of photon-return times directly, while analog lidars measure a time series of voltages that are approximately proportional to the rate at which photons are incident on the detector. Ideally, each of these types of measurement would give a good approximation of the time distribution of photons reflected from the ice, and a waveform measured by an analog lidar would be equivalent to a histogram in time of photons detected by a photon-counting lidar. In practice, the characteristics of the altimeter and the characteristics of the surface measured both play a role in the degree to which subsurface-scattering effects can be distinguished in the recorded waveform. In our model (see section 3), the waveform for a laser altimeter corresponds to the temporal convolution of the distribution of photon delays, the impulse response function (IRF) of the recording system, the range to the surface, and the shape of the transmitted pulse, so effects of subsurface scattering become easier to measure for narrower transmitted pulses, higher bandwidth recording systems, flatter surfaces, and smaller beam divergence values. The recent (post-2017) versions of the ATM transceiver offer good potential to measure scattering effects, because the temporal resolution of the system (corresponding to the receiver sampling interval and the pulse duration) is not large compared with the path delays predicted for green light reflecting from snow surfaces (Smith et al., 2018). Similar measurements have been made using the Land, Vegetation, and Ice Sensor (LVIS) (Hofton et al., 2008), but because of that sensor's longer pulse duration and infrared wavelength, we expect its waveform shapes to have only limited sensitivity to snow conditions. Photon-counting lidar measurements by the Slope Imaging Multi-polarization Photon-counting Lidar (SIMPL) (Yu et al., 2016; Harding et al., 2011) offer some of the advantages of ATM data, but used a photon-counting detection strategy that is not compatible with the processing methodology used in this study.

**Table 1. Dates and instruments for ATM measurements.**

| Campaign | Instrument | Dates processed |
|---|---|---|
| Summer, 2017 | narrow-swath | July 7 – July 24 |

| Spring, 2018 | narrow-swath | 3 March – 1 May |
|---|---|---|
| Spring, 2019 | narrow-swath | 3 April – 14 May |
| Summer, 2019 | narrow-swath , wide-swath | 4 September – 11 September |

ATM waveform measurements in this study come from data collected in Greenland in the 2017 summer campaign, the 2018 spring campaign, and the 2019 spring and summer campaigns. Most of the data that we processed (summarized in table 1) were collected using the ATM narrow-swath scanner, but we also processed wide-swath data from the 2019 summer campaign. For both scanners, the laser's incidence angle on a flat surface is approximately half the full scan angle, thus 15º for the wide swath and 2.5º for the narrow swath. Waveform data from these campaigns are distributed in the ILNSAW1B and ILATMW1B products (Studinger, 2018a, b), which provide digitized transmitted and received waveforms associated with each transmitted pulse. The waveforms have a temporal sampling of 0.25 ns, and are quantized at 8 bits, to produce digital values between 0 and 255. A variable neutral density filter in front of the receiver determined the optical throughput of the system, and was set to avoid digitizer saturation over snow surfaces. We considered using the near-infrared waveform data collected during the 2019 Greenland campaign, but found that the signal-to-noise ratio of these data was much lower than that of the green data, and that over coarse-grained surfaces, the infrared return was often absent even when the green waveform showed a clear return. Therefore, to obtain a consistent set of measurements, we focus our study on the green waveforms.

At the start of each ATM measurement campaign, waveforms were recorded with the laser aimed at a fixed, flat panel of fine-grained white material (Spectralon®) (Studinger et al., 2022a). We take these measurements to represent the system IRF $I(t)$ for the whole campaign. Although ATM instruments record both the received and transmitted waveforms, we found that the recorded transmitted waveforms were not a good representation of the system impulse response (see supplemental material section S1). Because of this, we disregard the measured transmitted pulse shapes, and instead assume that the system IRF is consistent with the most recent calibration measurement available. The wide-swath and narrow-swath ATM instruments produce very similar measurements, but use separate transmitters, optics, and receivers; for this reason, we use separate calibrations for the two systems for each campaign (Studinger et al., 2022b).

## 2.2 Grain-size estimates from the AVIRIS-NG airborne spectrometer

To help evaluate whether the ATM-derived waveforms were consistent with the returns we would expect from known surface conditions, we used data collected using AVIRIS-NG, on a separate aircraft that followed the aircraft carrying ATM on five subsequent days in the autumn of 2019. AVIRIS-NG measures radiances at 425 different wavelengths between 380 and 2510 nm on a detector array that produces images with 598 across-track samples (Thompson et al., 2018); its ~7.5 km altitude during the 2019 survey produced images on a ~4-5 km-wide swath, with ~6-7 m pixel sizes (Nolin and Dozier, 2000). These measurements were processed to estimate grain sizes using a technique that uses the strength of an absorption feature in the reflectance spectrum of snow at 1.03 $\mu m$ as an indicator of snow grain size (Nolin and Dozier, 2000). We rejected one of the

data files (the single file collected on 9 September, 2019, and the only file with extensive coverage of sea ice) because while the image appears to resolve a melting surface including a variety of sea-ice features including melt ponds and leads, the range of retrieved grain sizes span a small range ( 90% of values between 164 and 287 µm). The reason why this file should contain anomalous values is not clear, although we note that the sun was lower in the sky than it was for any other file (79º solar zenith angle, as compared to ~70-72º for other files in the campaign), which we hypothesize might result in lower-quality grain-size retrievals. The remaining 26 data files cover two coast-parallel lines and a few coast-perpendicular lines in northwest Greenland, spanning a range of grain size conditions from large-grained melting surfaces near the coast to fine-grained surfaces inland, and 17 of these overlapped with available ATM waveform files. Most (~80%) overlapping measurements within a 5-day window were collected within three hours of one another, and to limit how much the surface might have changed between one set of measurements and the other, we compare measurements between the two systems only if the differences between timestamps for the data files are less than 200 minutes.

## 2.3 Grain-size estimates from OLCI reflectance measurements

To demonstrate potential corrections for ICESat-2 height biases, we use a set of satellite measurements (Vandecrux et al., 2022b) derived from the OLCI instrument onboard the European Space Agency's Sentinel-3A satellite. OLCI provides surface-reflectance information for 21 spectral bands over a 1270-km wide swath with sub-kilometer resolution, giving sub-daily revisit times for Greenland during summer months. Images that were determined to be cloud free were converted to grain-size estimates by comparing estimated surface reflectances at 685 nm (far red, band 17) and 1020 nm (near infrared, band 21) with the output of a radiative-transfer model (Kokhanovsky et al., 2019). The result is a set of daily maps of Greenland, posted at 1 km, giving an estimate of the surface optical grain size for cloud-free areas of the ice sheet (Vandecrux et al., 2022a). Validation of these maps (Vandecrux et al., 2022b) against ground-based grain-size estimates derived from the infrared (1310 nm) reflectance of surface-snow samples collected at EastGRIP in northeast Greenland found that the OLCI-based estimates were systematically larger than ground-based estimates, but showed the expected decreases during snowfall events and increases during melt events. We compare ATM and AVIRIS-NG grain-size estimates with the OLCI-based estimates by bilinear interpolation into each daily grid; if measurements are marked as invalid in an OCLI map because of the presence of clouds, we derive an estimate based on the previous day's map under the assumption that the grain size had not changed substantially between the two days, and if the previous day's estimate is invalid, we reject the data point.

## 3. Methods

Work in this study is based on a model of how the measured time distribution of light reflected from a scattering surface depends on the properties of the surface and on the properties of the transmitted waveform (Smith et al., 2018). We partially validate this model by comparing its results with measured waveforms, and by tuning the parameters in the model, we estimate

surface grain sizes in Greenland, and use these grain-size values as a proxy for the degree of subsurface scattering to help predict subsurface scattering delays in ICESat-2 data.

## 3.1 Modeling return time distributions

We model light scattering in snow and firn based on a Monte-Carlo radiative transfer model for near-surface scattering
190   combined with an analytical extrapolation of the shape of the return for photons with long scattering delays (Smith et al., 2018). This model is similar to that used in other studies (Allgaier and Smith, 2021), except that we use a Monte-Carlo model to predict the return photon distribution at short delay times and diffusion theory at longer delay times, where the other studies use diffusion theory at all times. The choice to use diffusion theory is appropriate when the detector and the laser source are not coincident (i.e. when all photons measured have travelled an appreciable horizontal distance through the scattering
195   medium) but less so for the backscatter geometry used here, because diffusion theory can produce unphysical results for very short time delays (Flock et al., 1989). For measurements in which there is a horizontal offset of more than a few times the scattering length between the source and the detector, these short delays are not observed, whereas in the backscatter geometry of an altimetry measurement, many photons are likely to return after only a few scattering events. By directly modelling the time of flight for the incident beam and the first few scattering events, our Monte Carlo model avoids this problem.

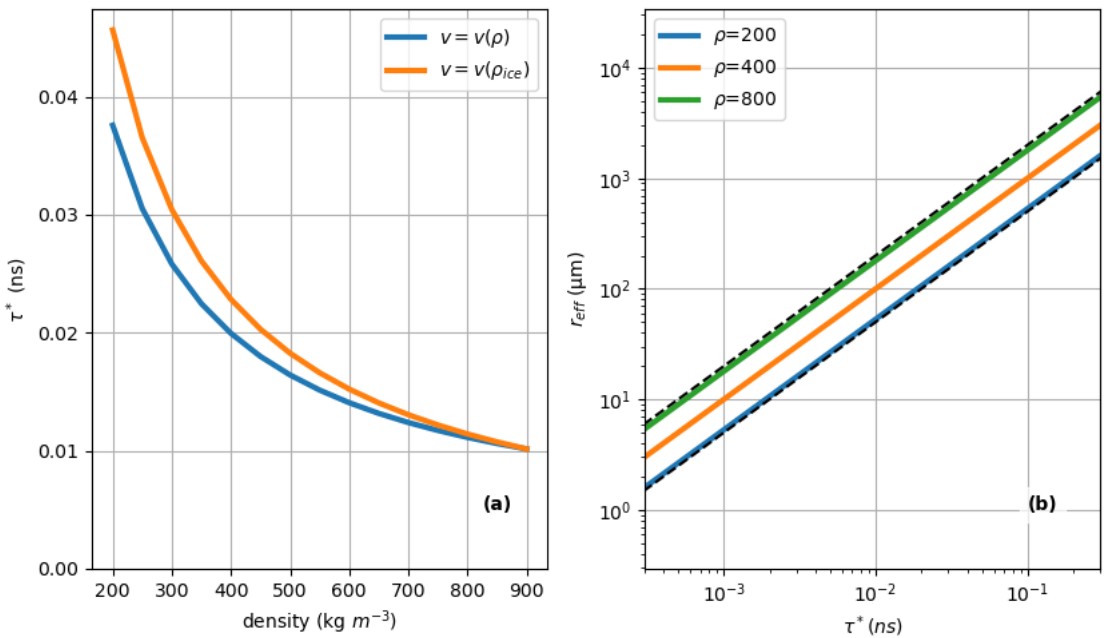

200

**Figure 1. Relation between scattering time, density, and effective grain size. Panel A) shows the relation between scattering time and density for a constant grain size of 200 μm, using a mixing law to calculate the velocity, and using a constant velocity appropriate to solid ice. Panel B) shows the relationship between scattering time and grain size, for three different densities. The dashed black lines show double and half the effective radius for $\rho = 400 \ kg \ m^{-3}$.**

205

Returns from our model can be described as:

$$SRF_m(t) = S\left(\frac{t}{\tau^*(r_{eff}, \rho)}\right) \exp\left(-k_{abs}(r_{eff}, \rho)v_{eff}(\rho)t\right) \tag{1}$$

where

$$\tau^* = \left(v_{eff}(\rho)k_{scat}(r_{eff}, \rho)(1 - g(r_{eff}))\right)^{-1}$$

Here $v_{eff}$ is the effective velocity of light traveling through the scattering medium, which depends on the density; $k_{scat}$ and $k_{abs}$ are the bulk scattering and absorption coefficients of the medium; $g$ is the asymmetry parameter of scattering in the medium, $r_{eff}$ is the optical effective grain size, corresponding to the radius of a collection of ice spheres that would have the same surface-to-volume ratio as the scattering medium (Grenfell and Warren, 1999), and $S$ is a scattering function that gives the distribution of return times from a non-absorbing scattering half space, in units of the average time between scattering events in the half space. The quantity $\tau^*$ describes the time required for light to travel between two scattering events, where we have approximated the anisotropic scattering characteristics of light interacting with large particles by multiplying the scattering coefficient by a factor $(1 - g)$ (Smith et al., 2018). We estimate the optical bulk scattering properties based on a Mie-theory calculation treating ice grains as independent spheres of ice surrounded by air (Gardner and Sharp, 2010), which gives estimates of $k_{scat}$ and $k_{abs}$, and $g$ as a function of wavelength, grain size, and density. We approximate the velocity of light in firn for density $\rho$:

$$v_{eff} = c\left(\frac{\rho}{\rho_{ice}}n_{ice} + \frac{\rho_{ice} - \rho}{\rho_{ice}}n_{air}\right)^{-1} \tag{2}$$

where $c$ is the speed of light in a vacuum, $\rho_{ice}$ is the density of ice, $n_{ice}$ is the real part of the refractive index of ice calculated from a published compilation (Warren and Brandt, 2008), and $n_{air} = 1$.

To reduce our description of scattering to a single parameter, we use a nominal density value of 400 kg m$^{-3}$, and a corresponding velocity value of 0.27 m ns$^{-1}$, which lets us express Eq. 1 solely in terms of $k_{abs}$ and $r_{eff}$. Although the choice of 400 kg m$^{-3}$ is somewhat arbitrary, it strikes a balance between the smaller, 270-350 kg m$^{-3}$, densities typical of Greenland snow (Fausto et al., 2018) and the larger, 410-910 kg m$^{-3}$ densities observed in melting snow and glacier-ice surfaces (Cooper et al., 2018). Figure 1A shows $t^*$ as a function of density for a grain size of 200 µm, plotted once using the relationship between velocity and density from Eq. 2, and once using a constant velocity value appropriate for solid ice. Over this range of densities, $t^*$ varies by about a factor of 4, while the difference in $t^*$ associated with the velocity model is at most about 20%. This shows that most of the variability in scattering time is associated with the distance between scattering events (determined by the density and the grain size), not with the velocity of light in the medium (determined by the density alone). Figure 1B shows grain size that would be inferred for a given $t^*$ value, for our nominal density value (400 kg m$^{-3}$), and for densities corresponding to light, fresh snow (200 kg m$^{-3}$) and to nearly solid ice (800 kg m$^{-3}$). Over the three-order-of magnitude range

of $r_{eff}$ considered here, the range of $r_{eff}$ at any given value of $t*$ between the nominal and the extreme values of density is just less than a factor of two, which demonstrates that while there is some uncertainty in the relationship between $t*$ and $r_{eff}$ when the density is unknown, a measured value of $t*$ can constrain the surface grain size to around a factor of two.

## 3.2 Modelling expected waveform shapes

The return waveform measured by an altimeter depends on the scattering properties of the surface, on the shape of the surface, and on the IRF of the system making the measurements. We calculate model surface-return shapes as:

$$W_{model}(t - t_{surf}, r_o, \sigma) = I_{est}(t) \otimes SRF_m(t; r_o) \otimes G(t, \sigma) \qquad 3$$

Here $W(t - t_{surf})$ is the received waveform, where $t$ is time and $t_{surf}$ is the round-trip travel time to the surface, and $\otimes$ represents a temporal convolution. $SRF_m(t; r_o)$ is calculated from Eq. 1, $I_{est}(t)$ is an estimate of the system IRF. We approximate the distribution of photon delays due to slope and surface roughness weighted by the illumination pattern of the laser as a Gaussian function, $G(t, \sigma)$.

Our approximation of the effects of slope and roughness follows studies that modelled satellite laser altimetry waveform shapes (Yi et al., 2005; Smith et al., 2019a). If we assume that the illumination pattern is represented by a two-dimensional Gaussian function with standard deviation $\sigma_b$ and the illuminated surface is represented well by a rough plane whose normal makes an angle $\phi$ with the beam direction, and that the roughness produces a Gaussian distribution of elevations relative to the plane with standard deviation $\sigma_r$, then the standard deviation of the Gaussian function, $\sigma$, should be equal to $\frac{2}{c}(\sigma_r^2 + \sigma_b^2 \tan^2(\phi))^{1/2}$ (Yi et al., 2005; Smith et al., 2019b). This means that more strongly sloping surfaces should produce broader returns, and that returns from the wide-swath ATM instrument should be broader than those from the narrow-swath instrument. ATM's 0.7-m footprint implies that over a flat surface smooth surface, $\sigma \approx 1$ ns for the wide (15-degree incidence angle) swath , or 0.2 ns for the narrow (2.5-degree incidence angle) swath. Note that because ATM uses a conical scanning mechanism, each scanner's beam will intersect a flat surface with an incidence angle equal to the scanner's off-nadir angle.

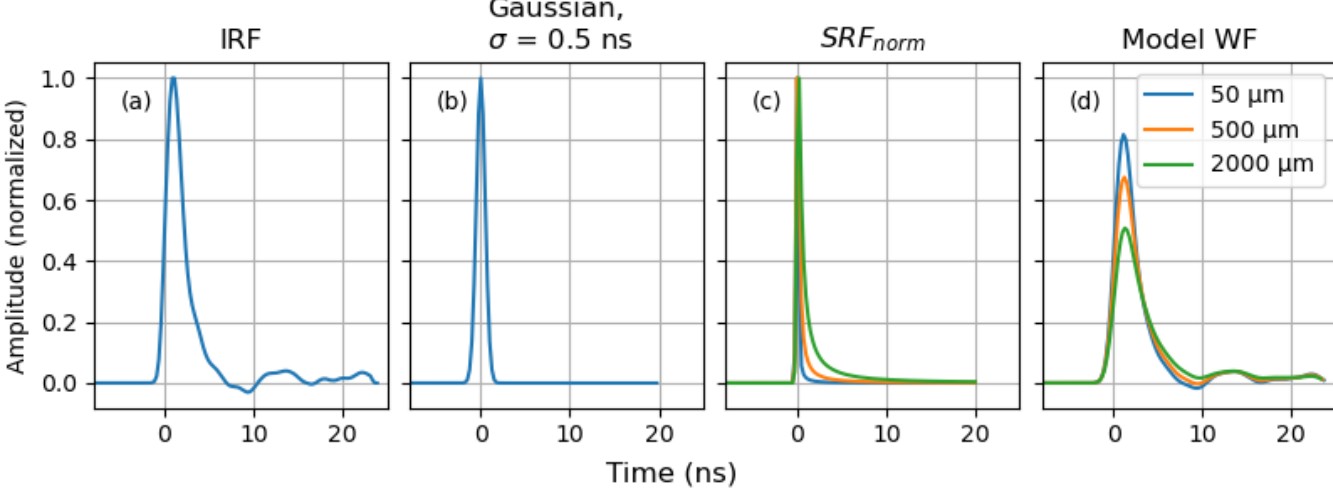

**Figure 2. Components of the waveform model. The ATM IRF (a) is convolved with a Gaussian function representing surface roughness (b) and the surface response function (c) to produce the model waveform (d). Three SRFs and corresponding waveforms are shown in (c) and (d), for $r_{eff}$=50, 500, and 2000 μm. Curves in (a-c) are normalized to unit amplitude, curves in (d) are based on an IRF with with unit amplitude.**

Figure 2 shows the components of Eq. 3, and resulting waveforms, based on the system IRF measured using a calibration target with no significant subsurface scattering on 9 March 2018, for a surface roughness equivalent to 0.5 ns (*i.e.* 7.5 cm), and for three snow grain sizes: 50, 500, and 2000 μm. The modeled waveforms show that for increasingly large grain sizes, the peak amplitude of the waveform becomes smaller and the waveform becomes broader, with the trailing edge of the waveform being blurred much more than the leading edge. The measured *I(t)* has a distinctive droop (negative excursion) just after the end of the main pulse, which is reflected in the predicted waveforms, although for larger grain sizes it no longer extends below zero. We were initially uncertain that the droop in the *I(t)* was due to a process that would be modeled correctly by Eq. 3, but the consistency between modeled and recovered waveforms (see section 4.1) suggests that the process that leads to the droop is a linear effect, likely in the receiver electronics. We speculate that it is due to bandwidth limitations in the receiver, perhaps due to an impedance mismatch at the input of the digitizer, but do not have strong evidence about its origin.

### 3.3 Matching modelled waveform shapes to measured waveforms

For each measured waveform, we identify the first sample at which the waveform exceeded 50% of its maximum amplitude and assume that all samples more than 3 ns before this sample contains only a uniform background offset and noise, whose values we calculate as the mean and standard deviation ($N_{est}$) of the sample values in this region. We then correct each waveform by subtracting this background offset.

To match waveforms with model results, we minimize the misfit between the DC-corrected and modelled waveforms:

$$R^2\left(r_{eff}, \sigma, t_0\right) = \sum\left[\frac{P_m(t_i) - A\,W\left(t_i - t_0, r_{eff}, \sigma\right)}{N}\right]^2 \qquad 4$$


Here $P_m(t_i)$ is the waveform sampled at times $t_i$, corrected for the background offset, and $W$ is the modelled waveform, $A$ is a scaling term relating the amplitude of the modelled waveform to that of the measured waveform, and $N$ is the number of samples in the waveform.

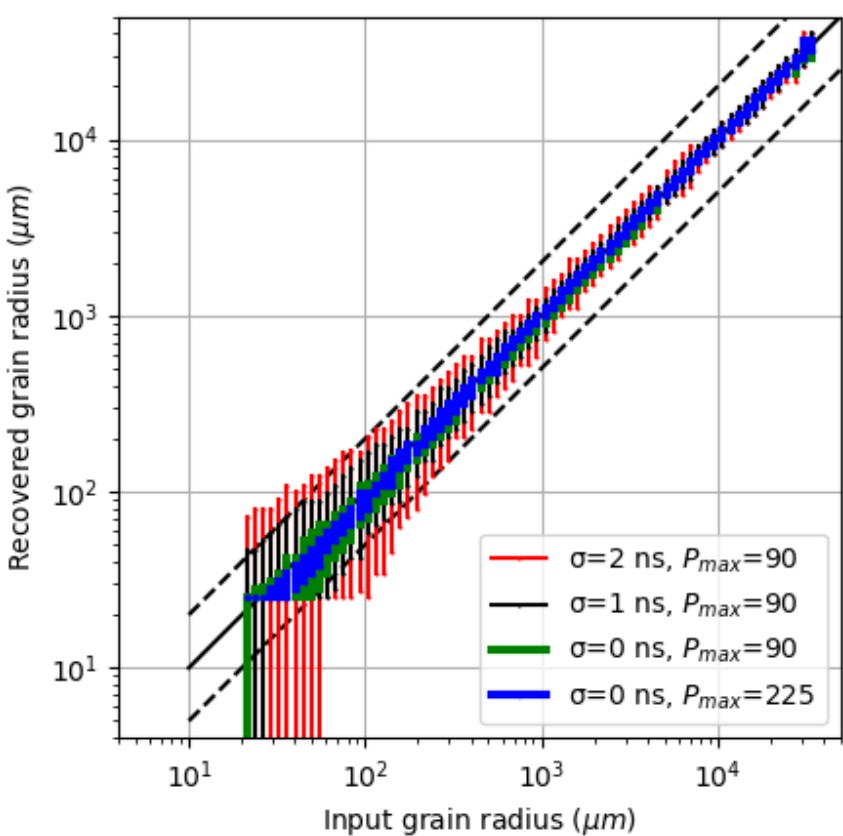


**Figure 3. Fitting test data. Vertical bars show the range of recovered grain sizes for each input grain size value, three low-amplitude, rough-surface cases with $P_{max}$=90 and variable pulse broadening ($\sigma$=2, 1, and 0 ns), and a high-amplitude case with $P_{max}$=255 and no pulse broadening ($\sigma$=0 ns). Bars indicate the 5th and 95th percentiles of the recovered grain sizes; bars extending off the bottom of the plot for the smallest grain sizes and the low-amplitude case indicate that for more than 5% of the waveforms, the best fit was**
**with the non-scattering model waveform. The solid line indicates a 1:1 relationship between the input and recovered grain sizes, and the dashed lines indicate the 0.5:1 and 2:1 relationships.**

We find optimal values for our adjustable parameters using a three-stage golden-section search (Press et al., 2007) in $\sigma$, $r_{eff}$, and $t_0$. The search algorithm consists of an outer search over a range of $r_{eff}$ values, with an inner search over $\sigma$ values, and

within that a second inner search over $t_0$ values. Within the search over $t_0$, the amplitude values are found with a least-squares

regression between each model waveform and the measured waveform. The searches use a tolerance in $\sigma$ of 0.25 ns and a logarithmic tolerance in $r_{eff}$ of 10%. After each golden-section search has converged, a final parabolic-search step is used to further refine the estimated $\sigma$, and $r_{eff}$ values. The convolutions in Eq. 3 are computationally costly, so we keep track of all waveforms we have calculated, and, whenever possible, use pre-computed waveforms in the misfit calculations. Using the golden-section search rather than a derivative-based searching strategy (*e.g.* a steepest-descent or conjugate-gradient search)

lets the fitting algorithm search a consistent set of parameters as it encounters waveforms that are similar to waveforms that it has previously matched, which greatly reduces the time required to fit a collection of waveforms, many of which are similar to one another. We further reduce our computational times by fitting only every fourth waveform for data from the narrow-swath scanner, and every second waveform from the wide-swath scanner. For most purposes in this study, we further reduce the spatial resolution of the recovered grain-size estimates using a 10-meter block-median filter, in which we identify the pulse

containing the median grain size value within each 10x10 m block sampled by each survey and report its location and grain size.

To evaluate the resolution and accuracy of this fitting procedure, we generated a set of test waveforms based on $I_{est}(t)$, for a range of grain sizes, pulse amplitudes, and broadening values. We assessed the sampling distribution of the recovered grain-size estimates by generating 256 different waveforms for each combination of parameters, normalizing each to a specified

peak amplitude ($P_{max}$), adding random (normal-distribution) values with a standard deviation of two digitizer counts to each sample, and applying our fitting algorithm to each. Our fitting algorithm selects grain sizes based on a set of pre-computed waveforms generated for grain-size values separated by 10%, so to demonstrate the worst-case performance of our algorithm, we generated the test data based on grain sizes that were half-way between the grain-size values used by the algorithm. Figure 3 shows the relationship between the specified and recovered grain size for small amplitudes and a range of broadening values

($P_{max}$ = 90, $\sigma$ =0, 1, and 2 ns), and for large amplitudes and small broadening values ($P_{max}$=225, $\sigma$ =0 ns). For the high-amplitude waveforms with little broadening ($P_{max}$=255, $\sigma$ =0 ns), the fitting procedure consistently recovers grain sizes as small as 20 μm, converging to either the next larger or the next smaller grain size value among the searched values (separated by 10%) with a moderate preference for the next smaller value, giving recovered values whose distribution width (5th to 95th percentile) is on the order of 10%. At smaller amplitudes ($P_{max}$=90) and larger pulse broadening values ($\sigma$=1, 2 ns), the width

of the recovered distribution increases with decreasing grain size, with the 5th and 95th percentiles of the distributions spanning around a factor of 5 for $r_{eff}$=50 μm and $\sigma$=2 ns. For input grain sizes up to about 75 μm (a factor of three times the minimum grain size tested), the waveform that best fit the simulated waveform was often the one with no scattering for the low-amplitude and broadened waveforms (A=90, $\sigma$=2 ns.) In these cases, the bottom of the distribution is not constrained on a log scale.

Our numerical experiments show that for synthetic data, the ratio between the amplitude of the pulse and the RMS of the noise added to the synthetic waveform plays a large role in the accuracy of the recovered grain size, with larger signal-to-noise ratios

corresponding to higher precision. For measured field data, the total gain of the system was set in advance using a neutral density filter to avoid detector saturation over snow surfaces, while the noise values were nearly constant, likely determined by the digitizer and receiver electronics. This should result in data with maximum signal-to-noise ratios over flat fine-grained snow surfaces, and lower signal-to-noise ratios over rough, sloping, and/or coarse-grained surfaces. Fortunately, the model results suggest that we should be able to recover grain sizes with small fractional errors when the grain sizes are large, even when the signal-to-noise ratios are relatively low.

### 3.4 Predicting biases in ICESat-2 measurements.

We predict expected biases in ICESat-2 data based on measured ATM waveform shapes by using our model to interpret the measured ATM waveforms, using the effective grain size as a proxy for the degree of subsurface scattering, then using the model again to estimate the range delay that would result from an ICESat-2 measurement over the same surface. To explain why this is necessary, we present a general statement of the magnitude of the bias (B) in an altimetry measurement estimated from a waveform $W_s(t)$, due to subsurface scattering:

$$B(M, W_s(t)) = M(W_s(t)) - M(W(t)) \qquad 6$$

Here $W_s(t)$ is the waveform including the effects of scattering, $W(t)$ is the waveform excluding the effects of scattering, and $M()$ is a metric used to derive height measurements from waveforms (referred to here as a *retracker*). The ICESat-2 ATL06 algorithm (Smith et al., 2019b) provides a standard land-ice height parameter, $h\_li$ that is based on the median photon elevation within a small (typically ±1.5 m) window around the surface. Ideally, to evaluate the expected biases in this parameter, we would use measured ATM waveforms to approximate $W_s(t)$, and use the ATM IRF to approximate $W(t)$, which would let us directly use Eq. 6 to calculate expected biases with the windowed waveform median as $M(\ )$. This is not practical, however, because most ATM waveforms include digitizer output that is less than zero (see Fig. 2). ICESat-2 uses a photon-counting lidar, so the median elevation can be calculated directly from the distribution of photon heights within the window. For a waveform lidar, the waveform median can be approximated under the assumption that waveform's digitzer counts (W(t)) are proportional to the flux of photons into the detector:

$$T_{med}(W(t)) = t | \frac{\int_{t_0}^{t} W(t')dt'}{\int_{t_0}^{t_1} W(t')dt'} = 0.5 \qquad 7$$

but if the relationship between the two is more complex (i.e. if I(t) in Eq. 3 is significantly different from a delta function), the waveform median may not be equal to the median time for the energy incident on the detector. This appears to be the case for ATM, where the recorded waveforms include negative values, implying a more complicated relationship between the photon flux and the recorded values.

Since we cannot apply the median retracker directly to the ATM waveforms, we model the effects of subsurface scattering on ATL06 biases by using Eq. 3 to generate synthetic scattering-affected waveforms for a range of grain sizes, based

on an estimate of the ICESat-2 system IRF derived from pre-launch calibration measurements (Smith et al., 2018). We then use Eq. 6 to predict the bias in the ATL06 measurements from each modelled waveform. Figure 4 shows the expected range bias for three retrackers as a function of grain size: the median retracker applied to the ICESat-2 IRF (the ATL06 *h_li* parameter), for a windowed mean on the same IRF (the ATL06 *h_mean*), and for a 15%-threshold centroid retracker (the metric used to track ATM waveforms), using the ATM IRF. The biases are smallest for the median retracker for the ICESat-2 waveform, increasing from sub-centimeter levels for $r_{eff}$ < 10 µm to around 35 cm for $r_{eff}$ > 10000 µm. The mean-based ICESat-2 bias is around twice as large as the median-based ICESat-2 bias, and the ATM bias is a few percent larger than the ICESat-2 median. This plot illustrates one difficulty in measuring ICESat-2 subsurface-scattering biases using laser-altimetry data as a reference: Over coarse-grained surfaces, ATM measurements are expected to have approximately the same biases as ICESat-2 measurements.

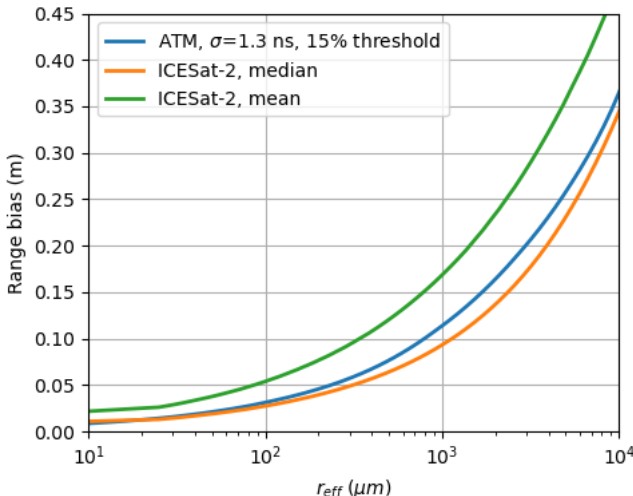

**Figure 4.** **Predicted range bias for ATM and ICESat-2 waveforms. ATM biases are calculated using a mean-based retracker with a 15% amplitude threshold. ICESat-2 biases are calculated using a windowed median and a windowed mean retracker.**

### 3.5 Subsurface-scattering-bias correction based on ATM and OLCI

To systematically correct ICESat-2 measurements, we need a spatially and temporally contiguous map of estimated subsurface scattering biases. In principle we could do this in two stages, using maps of grain size based on optical reflectance measurements (i.e. from OLCI), to interpolate a grain-size value for each ICESat-2 elevation measurement, and then calculating a range bias based on the relationship between grain size and bias using Eqn. 6. The accuracy of such a correction depends on the accuracy of the interpolated grain sizes and on the accuracy of the range bias predicted for each grain size. In

particular the accuracy of the predicted range biases depends on whether the same scattering processes that influence the range bias determine the surface reflectance, which may not be true in all cases. For example, OLCI and AVIRIS-NG grain-size

estimates are based in part the reflectance of infrared light (Nolin and Dozier, 2000; Vandecrux et al., 2022b), which, because of the stronger attenuation of infrared light by ice (Warren and Brandt, 2008), does not penetrate as far below the snow surface as green light does (Smith et al., 2018). As a result, the reflectance-based measurements may be more sensitive to near-surface layers than ICESat-2 would be. An ICESat-2 bias predicted based on surface reflectance measurements using our nominal 400 kg m$^{-3}$ will also be imprecise by up to a factor of two for snow and ice surfaces with smaller or larger densities.

385        In contrast to reflectance-based grain-size estimates, ATM-waveform-based grain-size estimates involve the same physical processes involved in ICESat-2 subsurface scattering biases. This implies that if we use the same model to interpret ATM waveform shapes that we use to predict ICESat-2 biases, the predicted ICESat-2 bias for a given recovered grain size should be consistent with the conditions that produced the ATM waveform, despite errors in the grain-size estimates related to surface-density variations. For this reason, we believe that we can use predicted biases based on ATM grain-size estimates

to evaluate bias corrections based on OCLI grain-size estimates, even if the OLCI and ATM grain-size estimates do not agree on a point-for-point basis.

The simplest way to calculate an OLCI-based correction to the ATL06 *h_li* parameter is:

$$B(x, y, t) = B_{med}(r_{OLCI}(x, y, t)) \qquad\qquad 8$$

Here B(x, y, t) is the estimated bias at position (x, y) and time t, $r_{OLCI}(x, y, t)$ is the grain size estimated from the OLCI data at the same location and time, and $B_{med}$ is the median ATL06 bias predicted using Eq. 6 and the ICESat-2 IRF. Based on our

assumption that subsurface scattering affects ATM waveforms in the same way it affects ICESat-2 photon distributions, we treat biases based on ATM grain-size estimates as representative of the biases that would affect ICESat-2 if it measured the surface at the place and time where ATM made its measurements. This lets us evaluate B(x, y, t) by comparing it against $B_{med}(r_{ATM}(x, y, t))$, the ATL06 bias estimated for the grain size measured by ATM at the same location. Thus, the statistics of $B_{med}(r_{ATM}(x, y, t)) - B_{med}(r_{OLCI}(x, y, t))$ should allow us to estimate the statistics of ICESat-2 ATL06 data corrected

using based on OLCI grain-size estimates.

As we will see in section 4.5, the OLCI measurements appear to become less sensitive to grain-size variations when the surface grain size is small. This leads us to also evaluate a threshold-based adjustment to the OLCI correction:

$$B_{thr}(x, y, t) = \begin{cases} B_0 & : r_{OLCI}(x, y, t) < r_{thr} \\ B_{med}(r_{OLCI}(x, y, t)) & : r_{OLCI}(x, y, t) > r_{thr} \end{cases} \qquad\qquad 9$$

Here $r_{OLCI}$ is the OLCI-estimated grain size, $B_{med}(r_{OLCI})$ is the model predicted bias, $r_{thr}$ is the threshold grain size above

which the model produces reliable bias estimates, and $B_0$ is a constant bias value used for OLCI grain sizes smaller than $r_{thr}$. We can use the distribution of recovered grain size values to find values of $B_0$ and $r_{thr}$ that minimize the mean and spread of $B_{med}(r_{ATM}(x, y, t)) - B_{thr}(r_{OLCI}(x, y, t); B_0, r_{thr})$.

## 3.6 Robust measure of spread

Throughout the results of this study, we will measure the width of distributions using the *robust spread*, which we define as half the difference between the 16[th] and 84[th] percentiles of a distribution. This is analogous to the standard deviation of a normal distribution, in which the central 68% of the distribution falls within one standard deviation of the mean. It allows us to characterize the spread of the central peaks of distributions that are not necessarily normally distributed, and for which the standard deviation might be dominated by large outlying values.

## 4. Results

### 4.1 Recovered snow grain sizes from ATM and AVIRIS-NG

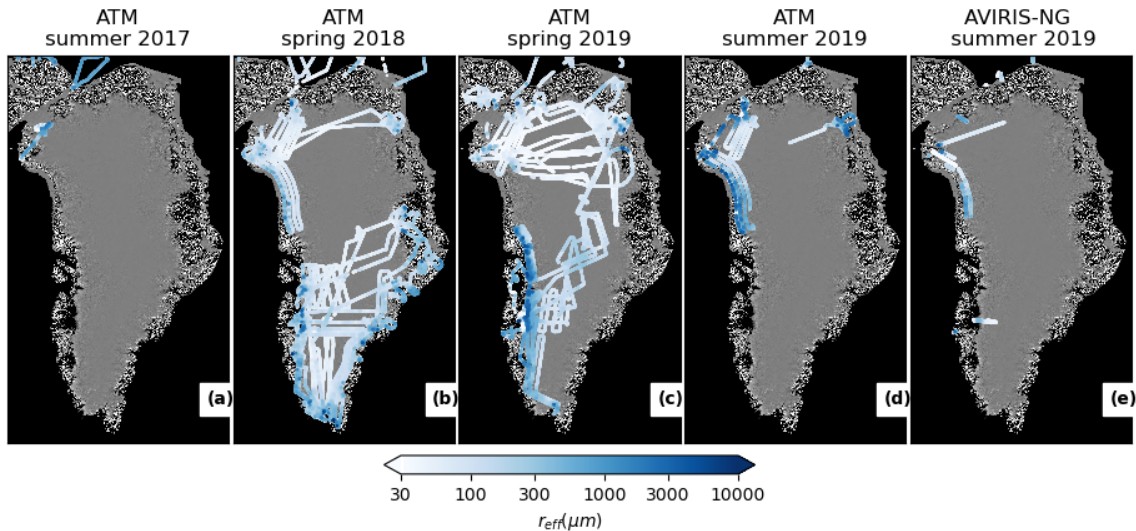

**Figure 5. Recovered snow grain sizes from ATM and AVIRIS-NG. Colored points indicate recovered grain sizes for four ATM campaigns (a-d) and for AVIRIS-NG (e). Each color-coded points indicates a 1-km block median of recovered grain sizes, and the**
**points have been plotted in order of grain size, so that coarser grain sizes overprint finer grain sizes. Background is the Mosaic of Greenland from 2015 (Haran et al., 2018)**

We obtained grain-size estimates from ATM for the summers of 2017 and 2019, and from the springs of 2018 and 2019. Figure 5 shows maps of recovered grain size from ATM and the valid AVIRIS-NG surveys for the late summer of 2019. These maps show a trend from large grain sizes at low elevation to small grain sizes at higher elevation, with notably larger grain sizes in
the summer than in the spring where surveys overlap. The southern portion of the spring-2018 survey (Fig. 5b) was carried out earlier in the season than the corresponding portion of the spring-2019 survey (Fig. 5c), and encountered finer grain sizes, particularly along the coast, while grain sizes in the northern parts of both of these surveys were consistently fine. The summer surveys in 2017 (Fig. 5a) and 2019 (Fig. 5d) both encountered coarse grain sizes, particularly in the coast-parallel lines in 2019. The AVIRIS-NG survey from 2019 (Fig. 5e) has most of its overlap with the contemporaneous ATM survey along two

coast-parallel lines, but a third coast-parallel line where ATM measured some of the coarsest grain sizes of the campaign was not covered.

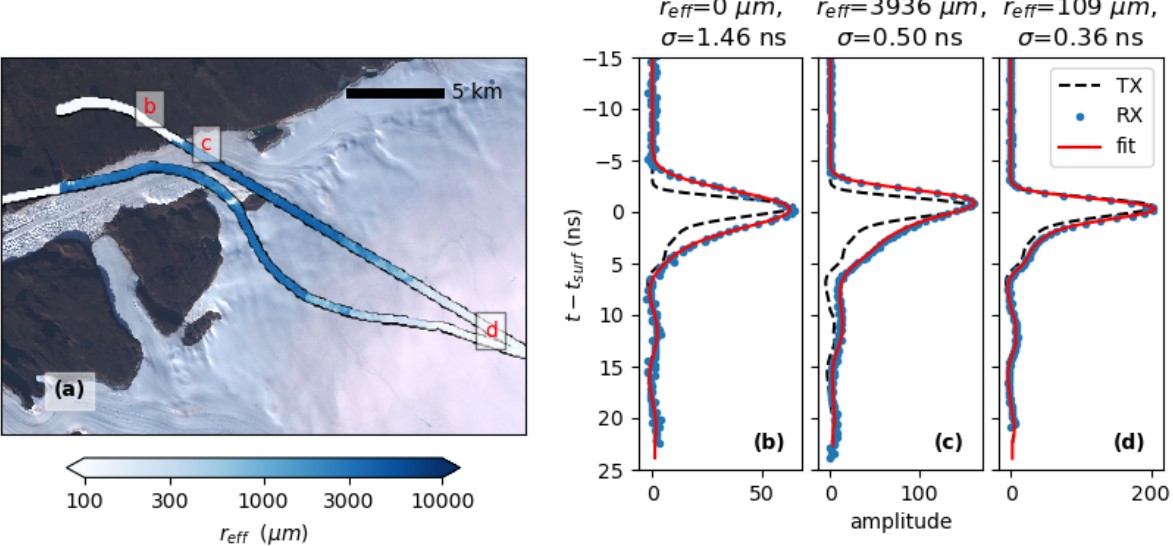

**Figure 6. Grain size and waveforms. (a) True-color (bands 4, 3, and 2) Landsat image of the Northeast Greenland ice sheet near**
**Leidy Glacier from 6 September, 2019, with estimated effective grain size (r_{eff}) from ATM data collected 4 September 2019. For the ATM data, we plot the results of a 100-m blockmedian applied to r_{eff}, and draw the outline of the swath in black. Panels (b), (c), and (d) show measured (RX) and best-fit modeled waveforms (fit), for three locations, as well as the input transmitted pulse (TX), scaled to match the amplitude of the received pulse. Bounding coordinates for panel (a) are presented in table S1. Landsat-8 image courtesy of the U.S. Geological Survey.**


To illustrate the spatial patterns of grain-size estimates recovered over a glacier during the melt season, Figure 6a shows a map of recovered grain size from Leidy glacier, northeast Greenland in the summer of 2019. We also show three waveforms, one measured over a rock/soil surface (Fig. 6b), one over low-elevation coarse-grained melting ice (Fig. 6c), and a third from finer-grained snow (Fig. 6d), as well as the corresponding best-fitting waveforms. The rock/soil waveform shows some

broadening relative to the transmitted waveform, likely due to surface roughness, that is symmetric in time, with equal distortion of the upper and lower slopes of the waveform. The best fitting model waveform has an $r_{eff}$ value of 0 $\mu$m, and a $\sigma$ value of 1.46 ns. The coarse-grained waveform (Fig. 6c) is also broader than the transmitted waveform, but has different amounts of distortion for the leading (upper) and trailing (lower) edges of the waveform: It has a sharply sloping upper edge, but a more gradual slope on the lower edge, which is consistent with the predicted effects of subsurface scattering. The best-

fitting model waveform has an $r_{eff}$ value of 2896 $\mu$m, and a $\sigma$ value of 0.26 ns. The higher-elevation waveform (Fig. 6d) has much less distortion than the low-elevation waveform, with a shape much more similar to the transmitted pulse, which is reflected in the best-fitting model parameters of $r_{eff}$ =109 $\mu$m, $\sigma$ = 0.26 ns. Elevations measured by ATM show that the outlet

section of the glacier (near (c)) is at 400-500 m, and elevation increases to around 1200 m near (d). The mapped distribution of grain sizes (Fig. 6a) shows little or no subsurface scattering on rock and soil ($r_{eff} \approx 0$), strong subsurface scattering for low-elevation ice ($r_{eff} > 1000\,\mu m$), and weaker subsurface scattering at higher elevations ($r_{eff} < 200\,\mu m$). We suggest that the lower-elevation part of the glacier on the left-hand part of Fig. 6a has experienced stronger surface melt than the higher-elevation part to the right, which is roughly consistent with the gradient from bluer to whiter tones in the background Landsat image collected two days later.

**4.2 Comparisons of recovered snow grain sizes between two independent ATM instruments**


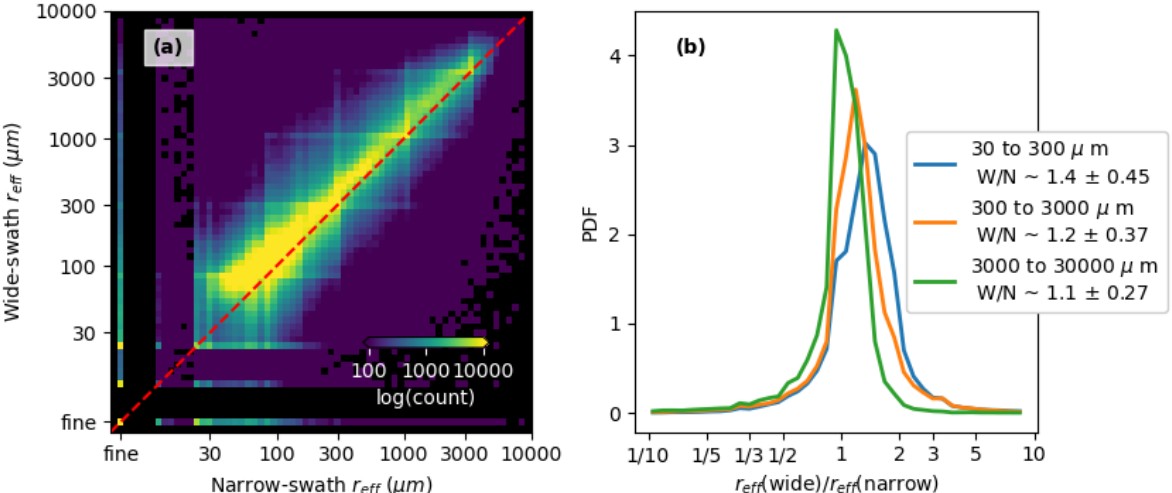

**Figure 7. Recovered snow grain sizes from two ATM systems from the Summer 2019 campaign. Panel a shows the density of measurements as a function of recovered r_eff values from the narrow and wide-scan ATM systems (lighter colors represent a higher density of measurements). Points for which one of the systems found a best match with a scattering-free model waveform are reported along the rows/columns marked 'fine'. Panel b shows the distribution of wide-to-narrow r_eff ratios for different ranges of narrow-swath r_eff. The legend for panel b gives the median and robust spread of the ratios for each range.**

Because the wide- and narrow-swath ATM instruments were installed on the same aircraft, there are abundant opportunities to compare measurements of the same surface at essentially the same time between the two, as a check on the self consistency of the measurements and as a check on whether the recovered grain size depends strongly on the incidence angle of the laser beam. Figure 7a shows a two-dimensional histogram of grain-size estimates from the wide and the narrow ATM sensors from the summer 2019 campaign. The estimates are clustered close to the 1:1 line, with slightly larger grain-size estimates from the wide-swath instrument. The histogram shows horizontal and vertical streaks that correspond to grain-size values that the

fitting algorithm selects preferentially as part of the effort to reuse previously computed model waveforms. These likely reflect small reductions in the accuracy of the recovered grain-size estimates, although not obviously to any large extent. For grain sizes smaller than around 25 µm, the fitting process for both datasets often selects a model waveform with no scattering model applied as best fitting the measurements. This results in a reduced number of recovered values at $r_{eff}<25$ µm, and spikes in the histogram for values where one or both estimates selected the scattering-free waveform. For display purposes, we have mapped

these to the left of and below the range of possible fit values (labeled 'fine' in Fig. 7a). The two sets of measurements appear to be consistent for grain sizes as small as 30 µm, and the two datasets report effective-zero grain sizes (< 10 µm) for most of the same points: for 85% of points for which the wide swath grain size effectively zero, the narrow swath was also, and for 70% of points for which the narrow-swath grain size was effectively zero, the wide-swath grain size was also.

The distribution of ratios between the recovered grain sizes for the two systems is similar to a lognormal distribution, with a central parameter close to unity. Figure 7b shows histograms of ratios between wide-swath and narrow-swath estimates, for three ranges of grain sizes (as determined from the narrow-swath values). For large grain sizes (> 3000 µm) the median ratio is 1.1, with a robust spread of 0.27; the bias and spread increase with decreasing grain size, and for small grain sizes (30 to 300 µm) the median ratio is 1.2, with a spread of 0.45. One possible reason for the larger grain-sizes estimates from the wide-

swath instrument is that the wide-swath beam had a larger incidence angle to the surface, so the return waveforms had somewhat larger Gaussian broadening. Our experiments with simulated data (section 3.3) suggest that 1 ns of pulse broadening can result in a small positive bias in recovered grain size for the 30-100 µm range of input sizes.

**4.3 Comparisons between snow grain sizes derived from ATM and AVIRIS-NG**

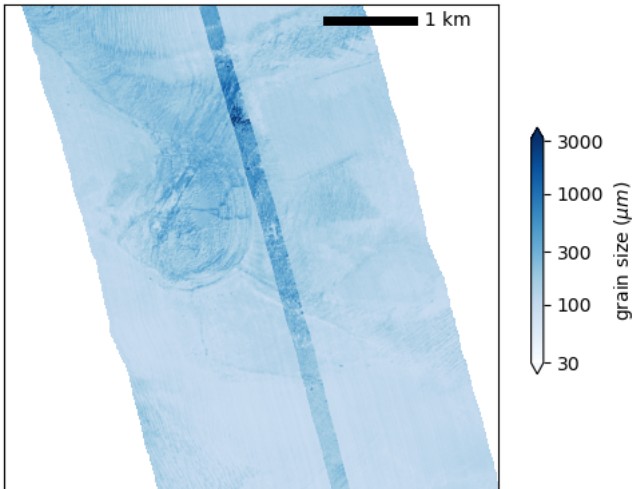

**Figure 8. Sample of AVIRIS-NG- and ATM-derived snow grain-size estimates for a coastal location in Greenland. The grain size based on the complete 4-km AVIRIS-NG swath is shown, with a 10-m block median of the recovered grain size from the 250-m wide-scan ATM swath superimposed on top. The scene center is approximately 75.314° N,**

**33.464°E, and contains data from the AVIRIS-NG granule ang20190906t144855 and the ATM granule ILATMW1B_20190906_133000.atm6T6.h5.**


Grain-size estimates from ATM and from AVIRIS-NG show consistent spatial variations, which are most easily identified in areas where the grain size varies on short spatial scales. Figure 8 shows maps of grain-size estimates from the wide-swath ATM scanner and from AVIRIS-NG for a short segment of a flight path in northwest Greenland. Both datasets show a range of surface grain sizes, with variations that that appear to correspond to spatial variations in surface weathering, likely over a

drained supraglacial lake basin. Despite the agreement between the small-scale variations, the ATM data show consistently larger gain sizes than AVIRIS-NG, particularly in the upper part of the scene in the roughest part of the lake basin.

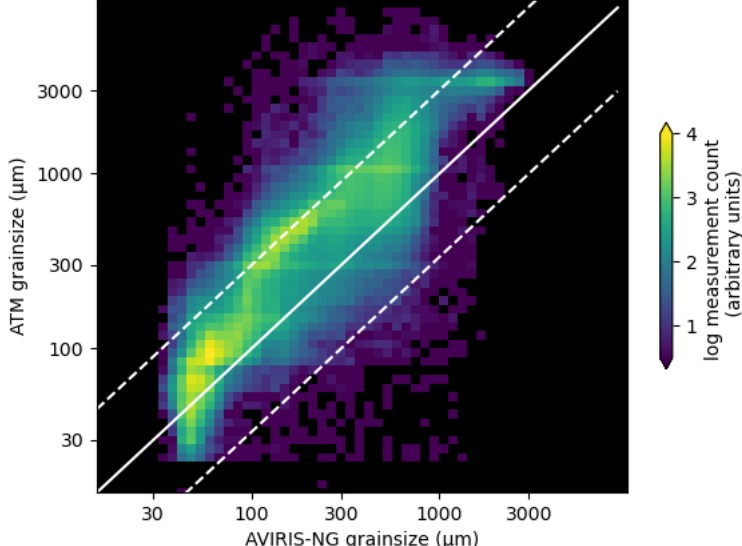

**Figure 9. Two-dimensional histogram comparing AVIRIS-NG-derived grain size with narrow-swath ATM-derived grain size, with cells colored by the number of points observed. The solid white line shows the 1:1 relationship between the two datasets. To help illustrate the magnitude of the difference between the datasets, we plot two dashed lines that show the ATM : 3 x AVIRIS-NG (upper) and ATM : 1/3 x AVIRIS-NG (lower) relationships.**

The general agreement between AVIRIS-NG and ATM grain-size estimates is illustrated by a comparison between 10-m

blockmedians of narrow-swath ATM grain-size estimates and AVIRIS-NG grain-size maps interpolated at the locations of the ATM data (Fig. 9). This plot was generated based on all narrow-swath ATM waveform data available for the ice sheet, but excludes a single AVIRIS-NG transect measured on sea ice, as discussed in section 2.2. For grain sizes greater than about 50 µm, the two show a generally similar trend, although ATM grain sizes are typically around 2-3 times larger than the corresponding AVIRIS-NG grain sizes. This relationship does not hold towards the small-grain size side of the plot, where

the AVIRIS-NG grain sizes are clustered in a near-vertical feature centered around 50 µm. We believe that this is because the

AVIRIS-NG algorithm loses some of its sensitivity to grain size variations around 40-50 μm while, based on our synthetic-data experiments, we expect the ATM retrievals to be sensitive to grain sizes as small as 25 μm. The points where the ATM fit selected zero scattering are not shown in this plot; they amount to a small fraction (0.4 %) of observations.

## 4.4 Comparison between OLCI, AVIRIS-NG, and ATM snow grain sizes

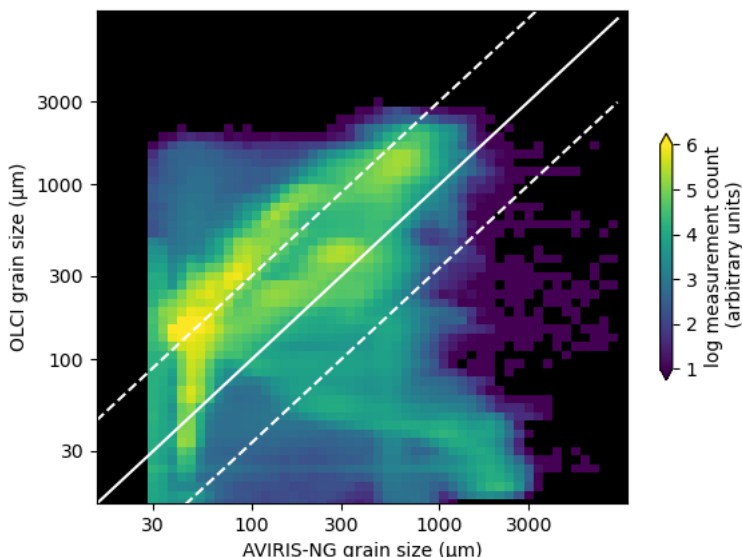


**Figure 10.  Comparison between AVIRIS-NG-derived grain sizes and OLCI-derived grain sizes.  The solid white line shows the 1:1 relationship between the two datasets, the two dashed lines show the OLCI: 3 times AVIRIS-NG (upper) and OLCI : 1/3 AVIRIS-NG (lower) relationships.  All OLCI measurements were collected within 1 day of the AVIRIS-NG measurement.**

Direct comparisons between the AVIRIS-NG and OLCI grain-sizes help illustrate the reliability of each dataset on its own and in comparison with ATM.  Figure 10 shows a 2-D histogram of AVIRIS-NG-derived grain sizes from the summer-2019 survey and OLCI-derived grain sizes collected within one day of the AVIRIS-NG measurements.  The largest concentration of OLCI grain sizes is between three and four times larger than the corresponding AVIRIS-NG sizes.  As in the comparison between ATM and AVIRIS-NG, there is a vertical feature in the distribution at AVIRIS-NG grain size = 40-50 μm, which likely
corresponds to the fine-grained limit of the AVIRIS-NG data.  The distribution of measurements for which the OLCI grain-size estimates are substantially finer than the AVIRIS-NG estimates may reflect contamination with undetected clouds in the OLCI imagery, which would tend to bias the OLCI estimates in the fine-grained direction.

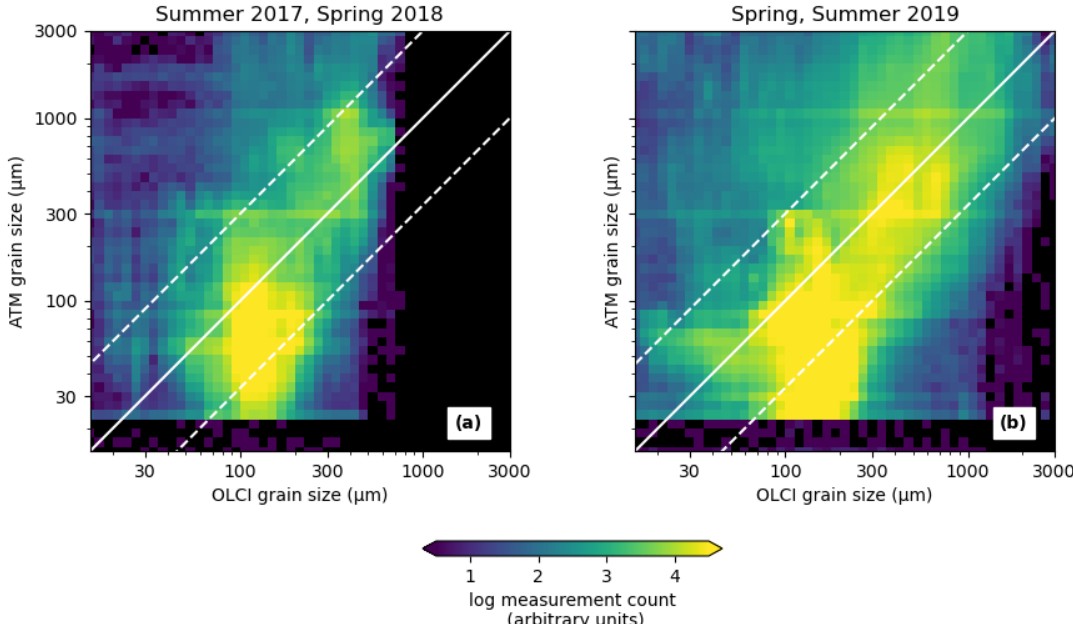

**Figure 11. Comparison between narrow-swath-ATM-derived and OLCI-derived grain-size estimates. Panel A shows**
**the distribution for the summer (July) of 2017 and the spring (March-May) of 2018, Panel B shows the distribution from the spring (April-May) and summer (September) of 2019. In both plots, the ATM grain sizes are derived from a 10-meter blockmedian of the data, and only those points for which the time difference between the OLCI measurement and the ATM measurement was less than 3 days are included. The solid lines indicates the 1:1 relationship between the datasets, the dashed lines indicate the 1:3 and the 3:1 relations.**

Similarly, Figure 11 shows a comparison between OLCI-derived grain sizes and those from the narrow-swath ATM instrument, based on a combination of data from the summer of 2017 and the spring of 2018 (Fig. 11a) and from the spring and summer of 2019 (Fig. 11b). In each case, the distributions of both types of grain-size measurements roughly follow the 1:1 line, although for both years, the ATM measurements show a range of measurements smaller than 100 µm for which the OLCI measurements are clustered around 100 µm. This may indicate that there are conditions under which the OLCI measurements

cluster around a moderately small grain size while ATM maintains sensitivity at smaller grain sizes. The 2017-2018 panel (Fig. 11a) contains far fewer points with large grain sizes because the dataset for the Summer of 2017 has very limited spatial coverage compared to the summer of 2019, and the Spring-2019 dataset covered more melting surfaces than did the Spring-2018 dataset.

**4.5 Comparing subsurface-scattering range bias estimates between OLCI and ATM data**


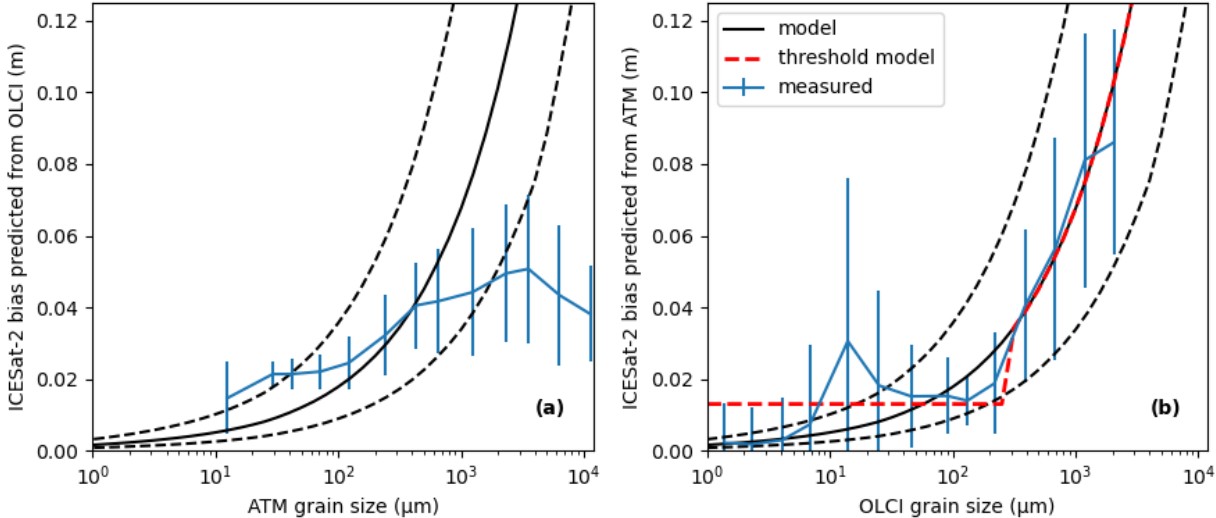

**Figure 12.  Range biases as a function of snow grain-size estimates for the complete 2017-19 dataset.  (a) shows range biases predicted from OLCI grain-size estimates as a function of ATM grain size, (b) shows range biases estimated from ATM grain sizes as a function of OLCI grain-size estimates.  For each panel, the vertical bars show the standard deviation of the range bias estimates for each grain size value, the black solid curve shows the modeled range bias as a function of grain size, and the dashed lines show the factor-of-two uncertainties in the model related to surface density. The dashed red line in (b) indicates the best fitting threshold model (Eq. 9).**

Comparing grain sizes estimated from the different sensors (Figs. 9-11) demonstrates the consistency (or lack thereof) between the datasets, but to address the usefulness of OLCI data in correcting biases in ICESat-2 data, we need to compare biases predicted for ICESat-2 based on OLCI with biases estimated based on ATM waveforms.  In these comparisons, the accuracy of the sensor is most important for large grain sizes because ICESat-2 biases predicted by our model (Fig. 4) are approximately zero for small grain sizes, so any correction we calculate will be small, with larger corrections expected for larger grain sizes.

If we assume that the ICESat-2 range biases predicted from the ATM waveforms are approximately correct, we can estimate the accuracy of OLCI-derived predictions of ICESat-2 biases in two ways: We can calculate the distribution of OLCI-derived predictions of ICESat-2 range bias for groups of ATM grain-size estimates (Fig. 12a), and we can calculate the distribution of ATM-derived predictions of ICESat-2 range bias for groups of OLCI-derived grain-size estimates (Fig. 12b).  In Fig. 12a , we collect groups of ATM grain-size estimates in logarithmic bins with a spacing of $10^{0.25}$ $\mu m$ and calculate the median and robust spread of biases of the ICESat-2 biases predicted from the corresponding OCLI grain sizes.  In Fig. 12b, we reverse this sampling and calculate the distribution of ICESat-2 biases predicted from ATM measurements for groups of OCLI-estimated grain sizes. In each set of axes, we plot the modeled relationship between grain size and range bias for reference.

The two plots in Fig. 12 cover different ranges of grain sizes because of the different ways that the two sensors sample the ice sheet. Fig. 12a includes large values of grain size from ATM (up to around 11000 µm) because single ATM measurements occasionally sample features on the surface with large grain sizes and includes no ATM measurements with grain sizes smaller than 10 µm because for smaller grain sizes, ATM often reports zero scattering. In Fig. 12b, grain sizes larger than 2000 µm do not appear, because the 1-km OLCI pixels rarely measure the small features where coarse grain sizes are observed. For the smallest OLCI-derived grain sizes, it appears that ATM often returned no-scattering estimates, so the estimated bias is effectively zero for both datasets.

On a per-ATM-waveform basis (Fig. 12a), OLCI bias estimates underestimate the sensitivity of ICESat-2 biases to grain size, especially for large ATM-derived grain sizes. This is likely because OLCI does not resolve small-scale coarse-grained features that are resolved by ATM (e.g. Fig. 8). In Fig. 12b, where the data are binned based on OLCI-derived grain size we see a closer match between the ICESat-2 biases predicted based on the ATM data and those predicted based on the OLCI measurements, at least for OLCI-estimated grain sizes larger than around 250 µm. At smaller grain sizes, the ATM-derived ICESat-2 bias estimates deviate from the OLCI biases, with a roughly uniform value close to 0.02 m for OLCI-derived grain sizes between 20 and 100 µm, a small peak for OLCI biases close to 15 µm, and approximately zero bias for finer grain sizes. This better correspondence shows that when OLCI-derived grain-size estimates can resolve coarse-grained features on the ice sheet, ATM measurements confirm the implied large bias values.

## 4.6 Calculating a best-feasible correction.

Based on Fig. 12b, it appears reasonable to believe that OLCI grain-size estimates provide useful information about subsurface delays for coarse-grained snow, but not for fine-grained snow. To better account for this lack of sensitivity in OLCI at fine grain sizes, we used the ATM and OLCI grain sizes from 2017 - 2019 to find optimal parameter values for the threshold bias model (Eq. 9): For a range of $B_0$ and $r_{thr}$, we calculated the median and the robust spread of the distribution of ATM biases corrected using on the OLCI grain sizes, $B_{med}(r_{ATM}) - B_{thr}(r_{OLCI})$. To help match the resolution between the ATM and the OLCI grain-size estimates, we carried out these calculations on a 250-m blockmedian of the ATM measurements. Figure 13 (a,b) show how the median and the robust spread depend on the parameter values. For threshold values greater than about 150 µm, there is a fine-grain-bias ($B_0$) value that gives a median residual of zero, and for each fine-grain bias, there is a threshold value that gives the minimum robust spread; these curves intersect at $B_0$=0.012 m, $r_{thr}$=270 µm. Figure 13c shows the distributions of ATM-derived biases, ATM-derived biases corrected based on $B_{med}(r_{OLCI})$, and of ATM-derived biases corrected based on the optimized $B_{thr}(r_{OLCI})$ model. The uncorrected distribution of ATM-derived biases has a peak at around 0.01 m a median of 0.013 m, with a substantial tail of values extending in the positive direction, representing coarse-grained parts of the ice sheet where we would predict that ICESat-2 would measure elevations several cm too low. Applying the unmodified correction results in a more compact distribution of residuals, with a median of -0.007 m and a spread of 0.006 m,

both of which are an improvement on the raw distribution but the bias is now in the opposite direction. The optimized threshold model yields a distribution of residuals with a zero median and a robust spread of 0.004 m.


The preceding analysis used robust statistics (i.e. the median and robust spread), which show how the correction works for typical locations on the ice sheet (i.e. ignoring the most extreme scattering conditions), which we would expect to fall in the middle of our distribution of residuals. However, many users of altimetry data will explicitly or implicitly perform their analysis using non-robust statistics (i.e. by calculating mean elevation differences, or calculating the standard deviation of

elevation differences). To show how the corrections work with statistics that are more sensitive to outlying values, we repeated the analysis using the mean and the standard deviation of the corrected datasets. This yields similar optimum $B_0$ and $r_{thr}$ values (0.014 m and 260 μm, respectively) for the zero-mean-residual model with the smallest standard deviation, but finds that for this model, the standard deviation is approximately the same as that for the non-optimized correction (0.011 m vs. 0.012 m). This shows that with the right parameters, the correction can produce a near-zero corrected mean, but cannot necessarily make

a substantial improvement in the standard deviation of the corrected data.

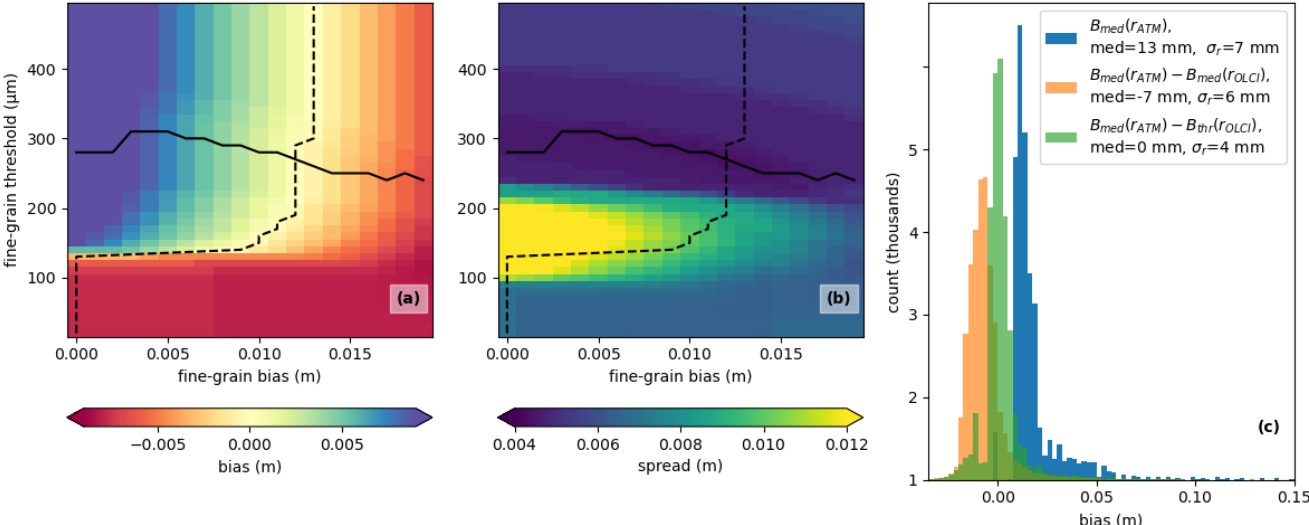

**Figure 13. Tuning the threshold correction for ATM-based ICESat-2 bias estimates. Panels (a) and (b) show the median and robust spread of the distribution of ATM-derived ICESat-2 bias estimates corrected with the threshold model (Eq. 9) for different values of the fine-grain bias ($B_0$) and fine-grain threshold ($r_{thr}$). The dashed curves show the fine-grain bias corresponding to the minimum**
**absolute value of the median for each value of the threshold, and the solid lines show the fine-grain threshold corresponding to the minimum value of the spread for each value of the fine-grain threshold. Panel (c) shows histograms of uncorrected bias estimates, bias estimates corrected based on $B_{med}(r_{OLCI}$, Eq. 8) and bias estimates corrected based on $B_{thr}(r_{OLCI}$, Eq. 9) for the optimum parameters, $B_0$=0.012 m, $r_{thr}$=270 μm. The median and robust spread of each distribution is given in the legend.**

## 5. Discussion:

The comparison of measurements between the narrow and wide-swath instruments (Fig. 7) shows that ATM-based estimates of snow grain size are consistent to within a factor of two or better between two independent instruments, and are not strongly

influenced by measurement geometry except at small grain size, where the larger angle between the wide-swath beam and the surface produces blurring of the returned waveform. Based on our modelling results (Fig. 3) and the expected relationship between incidence angle and return pulse width (Section 3.2), we expect this to result in larger scatter and bias in the wide-swath grain-size estimates. As estimates of grain size, the two sets of measurements have biases and uncertainties due to our assumptions about the density of the snow, but as measurements of photon delays due to subsurface scattering, they are reasonably consistent and should be useful in predicting biases in ICESat-2 data.

The comparisons between AVIRIS-NG and ATM grain size (Fig.9), and those between AVIRIS-NG and OLCI-derived grain size (Fig. 10) both suggest that the AVIRIS-NG estimates are biased by a factor of 2-3 towards fine grain sizes relative to the other dataset; further, both the ATM and the OLCI estimates appear to produce usable estimates of grain size that are smaller than 30 μm, while the AVIRIS-NG measurements seem to have a fine-grained limit of resolution around 40 μm. These differences between the AVIRIS-NG measurements and ATM-based measurements are consistent with comparisons between this AVIRIS-NG survey and observations of apparent elevation differences between green and near-infrared altimetry measurements that also implied that the AVIRIS-NG data had underestimated grain sizes (Fair et al., 2024). Despite these limitations, the comparisons between ATM, OLCI, and AVIRIS-NG measurements show broad agreement between the three sets of data, with larger grain sizes in each dataset corresponding to larger grain sizes in the others. However, this relationship is not as consistent as we might have hoped, and for a substantial fraction of the points there is no clear relationship between the grain sizes from the different sensors. Part of this scatter may result from differences in resolution between the datasets. ATM resolves grain size on a sub-meter-sized footprint, which we then degrade to 10 m using our blockmedian filter, the AVIRIS-NG data have a 5-meter pixel size, and the OLCI-based measurements are posted at 1 km. Many of the measurements showing the coarsest grain sizes from ATM are from small features such as crevasses and stream channels, which are likely not resolved by the larger pixel size of the OLCI measurements. Similarly, the smallest, coarsest-grained features in the AVIRIS-NG dataset are not expected to be resolved in the OLCI data.

There may also be differences between the retrieved grain sizes related to the measurement techniques. The ATM scattering measurements rely on subsurface multiple scattering that may sample hundreds or thousands of scattering events, and in which photons may penetrate hundreds of times the grain diameter below the surface. By contrast, the AVIRIS-NG and OLCI estimates both use portions of the reflectance spectrum extending into the near infrared, where the attenuation length of ice is as small as a few cm (Warren, 1982). This means that the ATM measurements are sensitive to grain size over a much larger range of depths than are the reflectance-based measurements. Particularly under melting surface conditions, we expect to see a layer of finer-grained ice on top of coarse-grained or water-saturated deeper layers (Cooper et al., 2018), which would lead us to expect that the reflectance-derived grain sizes would be finer than those derived from ATM. This effect is not expected to be as important under colder conditions, especially where fresh snow is present at the surface, because returns from a snow layer a few centimeters thick will contain only a very small minority of photons that have experienced long path delays (Smith et al., 2018)

We believe that it is also likely that there are disagreements between reflectance-derived measurements of grain size and ATM-based measurements because of the simplified relationship we have used between grain size and scattering properties. Our model of subsurface scattering assumes that the scattering is from independent spheres of ice suspended in air, and that the density of the medium is 400 kg m$^{-3}$. In fact, surface densities in the accumulation zone are often lower than that assumed by our model (Medley et al., 2022) while ablation-zone densities can approach that of compact glacier ice (800 kg m$^{-3}$ and higher),

and the presence of liquid water in the snow can result in reduced scattering efficiency per grain compared to that expected for spheres in ice. Over fresh, low-density snow, we expect our ATM-based measurements to overestimate grain size because our model does not fully account for the path length between scattering events and assumes that the extra path delay comes about because of time spent traveling through ice grains. Over compact ice surfaces the situation is more complex, because the surface density is likely larger than our reference density, leading to an underestimate of grain sizes, but close packing of grains

and the presence of water should each lead to less efficient scattering from each grain, leading to an overestimate of grain size. Under most circumstances, we expect the latter effects to be more significant, because the effect of density alone is unlikely to be larger than a factor of two (see Fig. 1).

The comparison between predicted ICESat-2 biases derived from ATM and those from the OLCI measurements (Fig.12) suggests that while OLCI measurements cannot accurately predict the measurement bias for each laser-based measurement,

the mean bias at the kilometer scale is more likely to be reliable. The difference between the two ways of plotting the biases as seen in Fig. 12 likely relates to the spatial resolution of the two sensors. ATM, with sub-meter resolution, captures small-scale features on the ice sheet, including crevasses, water channels, and ponds that all have large grain sizes. These features do not appear in the OLCI maps, which reflect the average grain size over 1-km pixels, which results in underestimates of bias for the ATM measurements with coarse grain sizes. Conversely, the average over OLCI measurements shows good agreement

with the predicted grain size-vs-bias curve, likely because the median biases for large, spatially distributed collections of ATM measurements are only weakly affected by the minority of ATM measurements collected over large-grain-size features. Further, the discrepancies between ATM and OLCI-derived grain sizes in the fine-grained regime (Fig. 11) should have relatively little impact on the accuracy of a OLCI-based prediction of biases in ICESat-2 data, because whatever their disagreements about grain sizes, the two datasets agree that the bias correction should be small. We hypothesize that the peak

in the ATM-bias-vs-OLCI-grain size plot around 20 $\mu m$ in Fig. 11b reflects undetected clouds in the OLCI data set; for these measurements, the ATM bias can have a large range of values, while the OLCI reports a grain size appropriate for polar clouds, resulting in an apparent positive shift in the ATM biases. Errors such as these might be ameliorated in part by combining reflectance-based grain-size estimates with a model of firn evolution, which might help identify unlikely values of grain size, but this kind of analysis is beyond the scope of this study.

Our experiments with a correction for ICESat-2 biases based on the OLCI-derived grain-size estimates (Fig. 13) show that for the full dataset, the mismatch between OLCI and ATM resolution and the imprecisions of the two datasets for small grain sizes result in a net overcorrection of the biases (shown in Fig.13c, where the median of the corrected range biases is less than zero) but a reduction in the spread of the corrected biases. Implementing a threshold-based simplification of the bias model that

assigns a constant value to the corrections for small grain size removes this bias and further reduces the spread of the residuals.

However, the optimum parameters of this threshold model are likely determined in large part by the characteristics of the input data, including the distribution of grain sizes included in the surveys and the accuracy of the OLCI grain-size estimates on the particular days during which each survey was conducted. Researchers interested in applying the same correction to a different set of satellite-based grain-size estimates would need to perform a similar analysis to calibrate the threshold values. To calibrate a new dataset of independent grain-size estimates against the ATM-based biases, researchers would need to repeat

the analysis that is summarized in Fig. 13:

1. Generate grain-size estimates for each ATM data point ($r_{est,sat}$)
2. Generate bias estimates for each grain size estimate ($B_{est,sat}$)
3. For a range of threshold values, calculate the median and spread of $B_{med}(r_{ATM}) - B_{thr}(r_{sat})$ (Eq. 9)
4. Select the threshold value that gives the minimum spread for a zero median

In our case, the threshold values that gave a zero median residual included those that gave a nearly optimal spread, but this would not necessarily be the case for other datasets, which would require more careful consideration of the trade-off between bias and spread in the correction. This kind of analysis is only feasible for satellite data that have temporal overlap with the existing ATM survey.

## 6. Conclusions

In this study, we have demonstrated a technique for the retrieval of ice-sheet surface grain size using the shape of pulses returned by a green-light laser. We showed that the shapes of the measured waveforms agree with the results of a simplified theoretical model of how subsurface scattering should affect the shape of green laser pulses, and experiments with synthetic data suggest that matching waveforms with the model results should allow accurate estimates of grain size over a wide range of conditions. We showed that measurements are consistent between two independent versions of the same instrument flown

on the same aircraft at the same time with different look angles, showing that the grain size recovery is repeatable, and is not strongly sensitive to the geometry of the measurements, except at small grain sizes for which the larger incidence angles associated with the wide-swath scanner begin to degrade the sensitivity of the system. Comparisons with reflectance-based estimates of grain size show agreement between the trends in the data, but not especially close point-for-point agreement between the ATM measurements and the reflectance-based measurements. However, comparisons between different

reflectance-based measurements also do not show point-for-point agreement, and we are unsure whether we should claim to have validated the novel ATM-based measurements with the better-established reflectance-based techniques or whether we should claim that our ATM-based measurements provide relatively precise ground truth for the reflectance-based measurements.

Returning to the original goal of this study, which was to predict biases in ICESat-2 data, we feel that the close agreement between ATM waveforms and the shapes predicted by our model validates our use of the model to predict ICESat-2 biases

due to subsurface scattering. The widespread large grain sizes we estimate in the low-elevation parts of Greenland suggest that there are large areas of the ice sheet for which we can expect decimeter-scale biases in ICESat-2 data. To date, our efforts to identify subsurface scattering bias in ICESat-2 data have been stymied by the need to collect data from tens or hundreds of
pulses to resolve the shape of the return waveform, which is difficult over the rough surfaces typical of low-elevation Greenland in the summer. This suggests to us that routine correction of ICESat-2 data based on ICESat-2 return-pulse characteristics will not be feasible, except perhaps for limited areas with unusually flat topography. However, the synthesis of the ATM and OLCI-based predictions of scattering delays (Figures 12b, 13) suggests that a correction based on satellite-derived estimates of grain size is feasible for the large grain sizes where biases are largest, and that an empirical adjustment of the relation
between grain-size estimates and predicted biases can be used to find a correction that yields an unbiased estimate with smaller variance than either the raw predicted biases or the unmodified correction model. Improvements in satellite-derived and model-derived estimates (Mei et al., 2021; Painter et al., 2009) of snow grain size are a potential way to improve the precision of a correction of this kind. One avenue for improvement might be to derive grain-size estimates from satellites with resolution finer than the kilometer-resolution OCLI data used here. A similar correction using Landsat and/or Sentinel-2 data could
provide data at 30-meter resolution, although with coarser time resolution and with a less optimal selection of spectral bands. Another possible data source for corrections of this type would be grain size predictions driven by a grain size-evolution model driven by meteorological data, remote-sensing data, or model output. Unlike grain-size estimates derived purely from satellite measurements, these would not be limited by the availability of cloud-free observations, and might be able to integrate remote-sensing data from multiple sources to reduce the effects of measurement errors. Any such comparison would require careful
consideration of the relationship between physical grain size (calculated in the grain size model) and the effective grain sizes considered in our scattering model, which might best be handled by calibrating model output overlapping the Greenland ATM surveys against ATM data.

**Data availability:**

ATM waveform data are available from the National Snow and Ice Data Center (Studinger, 2018a, b). Ground calibration data
used to derive the ATM instrument response is available at: https://zenodo.org/record/7225937. OLCI-based grain-size estimates are available through GEUS dataverse (Vandecrux et al., 2022a) AVIRIS-NG grain-size estimates are available by FTP from https://popo.jpl.nasa.gov/avng/y19/, and ATM-based grain-size estimates are available from the National Snow and Ice Data Center (NSIDC): https://doi.org/10.5067/1207YUVC7KOO.

**Author Contributions:**

BS developed the concept for the study and carried out most of the computations. MS participated in data surveys. TN, TS, ZF and MS contributed to refinement of techniques and analysis. TS helped with software development. All authors contributed to the draft preparation, review response, and editing.

**Acknowledgements:**

Work by BS was supported by NASA Cryospheric Sciences grants NNX17AI62G and 80NSSC20K1064. Work by TS was supported by NASA Cryospheric sciences grant 80NSSC22K0379. Work by TN was funded by the ICESat-2 project, and work by MS was funded by NASA's Internal Scientist Funding Model, the Airborne Science Program, and the Cryospheric Sciences Program. The authors thank the ICESat-2 project and the operation IceBridge project and flield teams for making this work possible and thank Steven Warren for initial inspiration of the study and David Harding for initial demonstrations of its feasibility. The authors would like to thank editor Louise Sørensen, referee Johnny Ryan and a second anonymous referee for their patience and hard work bringing this study to a finished form.

**Competing interests:**

At least one of the (co-)authors is a member of the editorial board of The Cryosphere.

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
