# Peer review of "Understanding biases in ICESat-2 data due to subsurface scattering using Airborne Topographic Mapper waveform data"

_The Cryosphere, 2023_

## Referee Comment (RC1)

tc-2023-147

*General Comments*

This manuscript presents a study aimed at predicting biases in ICESat-2 surface elevations due to the subsurface scattering of photons. The authors base their approach on the comparison of airborne laser waveforms acquired with the Airborne Topographic Mapping (ATM) system and grain size estimates from the Airborne Visible/Infrared Imaging Spectrometer – Next Generation (AVIRIS-NG) instrument and the Ocean and Land Colour Instrument (OLCI) on-board Sentinel-3. A subsurface scattering model is employed to derive grain size estimates from the ATM waveforms, which are then compared to the AVIRIS-NG and OLCI observations. Finally, these results are to predict possible biases in ICESat-2 elevations due to estimated grain sizes.

Overall, I found the manuscript very confusing and difficult to read. Many of the comments below will speak more to this point, but there is general a lack of specificity and clarity in how the manuscript is written. I think much of the confusion with regards to the scope of the manuscript comes from how it is initially presented as "Understanding biases in ICESat-2 data …" when there is no actual ICESat-2 data used. With how often ICESat-2 is mentioned in the Abstract and Introduction, the authors build an expectation that at some point what is discussed will be applied back those data, but it never actually happens. Also, as is stated on line 586, to date no biases in ICESat-2 due to subsurface scattering have been observed, which brings into question the overall relevance of the study. Are the authors proposing a solution to a problem that doesn't exist? The broad-scale importance of this work is not clear.

I think the manuscript would benefit from a thorough revision focused on improving context and clarity. I believe the foundational elements (i.e., approach, data analysis, results) of the study are interesting and worthy of publication, but how they have been packaged makes the full arc of the study difficult to follow. I also found the manuscript to be under-referenced. The authors often make direct statements without providing any supporting evidence.

I recommend the authors consider the following points while revising their manuscript. I realize there are a lot of them, but I want to be detailed and clear with regards to where I had trouble following the manuscript to help the authors with their revisions.

*Specific Comments*

- Abstract: Please provide more context for the last sentence. Why is it important to correct for subsurface scattering biases in ICESat-2 data? Within the context of the overall ICESat-2 height uncertainties, how big of a problem is it if these biases are left unaccounted for?

- Line 31: Please consider including the corresponding wavelength for ICESat to support the comparison to ICESat-2's 532 nm green laser.

- Line 47: Here weaker scattering is implied as being associated with broader returns and larger delays, but wouldn't weaker scattering be equivalent to stronger absorption in the medium? In the second and last sentence of the paragraph, stronger absorption is associated with tighter photon distributions.

- Line 50: Someone approaching the manuscript from an ICESat-2 (i.e., individual photon) perspective may be unfamiliar with the concept of a laser "waveform". I'd suggest the authors early in the manuscript define what a waveform is. Here a "return photon timing distribution" provides an excellent opportunity.

- Lines 70-72: The process of using a model for the shape of ATM waveforms to predict grain size and comparing that to airborne and satellite measurements is well laid out. What I continually struggle with is the subsequent extension to ICESat-2. What does it mean to "confirm" that predicted biases in ICESat-2 are the same as those for the ATM case when there are no actual ICESat-2 biases to compare against? Is the authors goal to show 1) that their scattering model is valid when using airborne (ATM and AVIRIS-NG) data, 2) can be applied to Sentinel-3 gain sizes to predict a generic laser bias, and 3) that generic bias compares favorably against ATM data, so that, in principle, one could possibly apply it to ICESat-2? So much of the emphasis in the manuscript is placed on applicability to ICESat-2 but by the end, that link seems like more of an inference/prediction/extrapolation than something that is concretely demonstrated (i.e., confirmed). Furthermore, I suggest the authors be more specific with what they mean by the "sampling" of altimetry measurements and how that affects calibration. What type of "sampling" are the authors referring to?

- Section 2: I know maps are presented in Figure 5, but I would recommend the authors consider including a composite overview map when describing the datasets to help situate the reader.

- Line 96: For those unfamiliar with the ATM data, please provide some specific detail as to what wide-swath and narrow-swath means (e.g., cross-track look angle, width on ground, etc.).

- Line 105-108: Here the authors refer to some analysis they have done that underlies the choice of impulse response function but isn't presented in the manuscript. I would recommend the authors consider including this so the reader can understand why these choices were made.

- Section 2.2: The authors include a lot of detail regarding the ATM system but almost none for the AVIRIS-NG system. How does this system operate? What does it measure? How big is the field-of-view? How closely did the Basler follow the aircraft with the ATM? I believe these details will help to provide context and clarity for the reader.

- Line 119: Please provide some example numbers for how much wider the AVIRIS-NG swath is compared to the ATM swaths.

- Line 122: The authors refer to AVIRIS-NG data with grain sizes inconsistent with the expected surface conditions that were not included. I would recommend being more specific regarding what evidence exists to suggest that these measurements are erroneous. Otherwise, it could leave the impression that the included data were cherry-picked.

- Line 138: I find this description of the "look-up table" and especially the "time steps" confusing. Can the authors simplify to something along the lines of "<time-specific> grain sizes are interpolated from the daily maps."? It is also not clear the degree of interpolation the authors are performing using the daily maps. Are the authors interpolating at sub-daily intervals or just over missing time periods due to clouds?

- Line 145: What assumption are the authors referring to here and why does in only apply to "most" of the data used in the study? What about the remainder?

- Section 3: I think there is something missing from this section, specifically how $\tau^*$ related to $r_{eff}$. $r_{eff}$ does not appear in any of the provided equations so it makes it very hard to follow the development of the waveform model as well as understand Figure 1b.

- Figure 1a: I am confused by Figure 1a. What is the point of introducing the constant velocity (orange) line? It seems to be to try and communicate the how sensitive $\tau^*$ is to the effective velocity of the medium, but in doing so does it not introduce a physically unrealistic scenario? That is, one where density is fixed as it relates to velocity but still allowed to vary with respect to the optical bulk scattering properties. How do the authors justify ignoring the density sensitivity in one variable in the $\tau^*$ equation, while keeping it in the remainder?

- Line 185: Could the authors provide some justification for their choice of 400 kg/m$^3$ as a nominal surface density value, when in-situ measurements across Greenland suggest a nominal surface density of 315 kg/m$^3$ (Fausto et al., 2018; https://doi.org/10.3389/feart.2018.00051)?

- Line 186: $r_{eff}$ does not appear in Equation 1. Please clarify as to where this is coming from.

- Equation 3: I recommend the authors provide more clarification on the difference between Equations 3a and 3b. It seems 3a is something akin to an "ideal" model and 3b is how the authors approximate it in this study. If that is the case, is it necessary to have both? What does 3a add that can't be communicated by 3b?

- Equation 3: I recommend providing an equation or a more in-depth discussion for the *G*-term in Equation 3b and how it has been selected. How is the surface roughness parameterized within this term? Is it an analytical or empirical representation of roughness? Has this type of parametrization been used previously? Over what roughness range is the model valid? What does it mean to have surface roughness expressed in time, as is done in Line 214? Without this detailed information, the reader cannot make any inference on whether the specific roughness model included in the model is applicable in this application.

- Equation 4: Please provide some explanation for what the *A* and *N* terms in this equation are and what they mean.

- Lines 257-259: I don't know what this sentence is trying to communicate. I don't understand how the test data have been generated or what "… half the spacing between the searched

values." is supposed to mean. I recommend the authors revise this sentence to make it clearer exactly what they have done.

- Line 270: What is the noise value the authors are referring to here? Please be more explicit in what it is and how it comes about.

- Lines 270-278: Much of this paragraph is dedicated to describing the effects ATM amplitude had (past-tense) on the uncertainty in the estimated grain size. The issue I find however, is that the ATM data results have not been covered yet. What do the authors expect the reader to take-away from this paragraph when they have not seen the grain size estimates from the ATM data yet? Furthermore, what does it mean for a surface to be "dark" (Line 275) with respect to laser altimetry? I suggest the authors elaborate or clarify this point. Finally, looking back on this paragraph after reading through the full manuscript, I find it odd that the discussion of precision or uncertainty was not carried through to the actual data analysis. Can the authors quantify the uncertainty in the grain size estimates produced from the analysis of the ATM data that they mention here?

- Line 290: Please consider elaborating further on why negative numbers affect the waveform-median retracker. Is there no other type of retracker that is not sensitive to sensitive to negative numbers that could be used? Also, I would recommend the authors consider including more detail on the two types of retrackers applied to the simulated ICESat-2 data (windowed mean and windowed median) for those who are not familiar with these specific ICESat-2 details. How do they work? The authors are assuming the reader is familiar with this nuanced part of ICESat-2 operations.

- Line 293: What is the IRF function for ICESat-2 and where do the authors get it from? I recommend the authors provide more elaboration on this point. Substantial space was given to establishing the ATM IRF in Section 2, while here the ICESat-2 IRF is almost glossed-over.

- Section 3.4: I think a key point I'm struggling to understand is the link between waveform model of Equation 3 and what is measured by a photon-counting laser altimeter such as ICESat-2. It underlies this entire section and the Section beginning on line 452 but it is not clear to me how this works. Perhaps including an example diagram of how the authors extract the more ICESat-2 relevant parameters from the modeled waveforms would help with this?

- Figure 5: I suggest the authors consider including specific dates in the titles of the subfigures instead of the generic "spring" and "summer" labels. The authors refer to earlier and later spring campaigns in the main body (Line 326) and including actual dates in the figure would make this clearer. Also, some of the detailed patterns the authors discuss in the main body (e.g., Line 331) are very difficult to see in Figure 5. I suggest the authors consider including a zoomed in version of the larger maps that highlight exactly what they are talking about.

- Line 326: When speaking to specific sub-figures, I recommend the authors actually refer to them (i.e., Figure 5b). This helps to make things much clearer for the reader.

- Figure 5 and Figure 6a: Please include coordinates for all maps.

- Figure 6a: The Blues colormap used to represent estimates grain size on top of the Landsat image is very difficult to distinguish. I recommend the authors consider a different colormap

that stands out more from the background. Another option would be to segment the grain size estimates based on the ranges presented in Lines 250-251.

- Line 350: Here the authors refer to the lower portion of the Leidy glacier as having experienced extensive melting. I recommend the authors include some justification for this as the distance between the larger grain size lower portion and the finer grain size upper portions is not much. Has extensive melting been so highly concentrated in only the lower section?

- Figure 7a: Please include a colorbar as is done for Figures 9 and 10.

- Figure 7b: Is there a specific reason as to why the distributions are presented on a log-normal scale? What are the units for the spreads provided in the legend? It seems odd to plot the data on a log-normal scale (especially something like a ratio) and then use the standard deviation. I recommend the authors explain why they expect the ratio between the wide and narrow swath ATM grain sizes to be logarithmically distributed.

- Line 363: To me "around" does not reflect the situation presented in Figure 7a. It appears as if the wide swath grain sizes are consistently larger than the those from the narrow swath. Perhaps it would be more representative to use a term like "near"?

- Figure 8: Please provide coordinates for the plot. Also, the authors mainly discuss the upper half of this plot. So would it prudent to zoom into the upper half in order to see the spatial patterns the authors discuss more clearly?

- Line 398: Here the authors refer to the comparison of the ATM and AVIRIS-NG gain sizes in Figure 9. I recommend the authors clarify which ATM dataset is being compared. Is it the narrow swath, the wide swath, both?

- Line 401: Here the authors refer to ATM grain size being equal to or smaller than the AVIRIS-NG grain sizes when the pattern look more like a loss in sensitivity in the AVIRIS-NG results at small grain sizes (the authors also state this on Line 494). Is this the most appropriate way to describe the results, as being equal to or smaller than? It implies that there is closer agreement between the datasets, whereas it appears more likely to be a numerical artefact.

- Figures 7a, 9, 10, and 11: All these plots show the same type of comparison between grain sizes. As such, I'd recommend the authors consider standardizing them such that the axis limits and colorbars are the same.

- Figures 9-11: I recommend the authors elaborate on why different bounds were chosen for the dashed white lines in these plots (i.e., factor of 2 in Figure 9 and factor of 3 in Figures 10 and 11). Also, please ensure the bounds are plotted properly as they look very similar between the factor of 2 and factor of 3 cases.

- Line 426: Here the authors state that comparison of grain size estimates "… roughly follow the 1:1 line.". Would it be possible to strengthen this argument using a quantitative value such as the correlation coefficient or some other metric?

- Lines 442-444: I find this sentence very confusing, and I don't know what the authors are trying to communicate. Why is it important to use OLCI grain sizes to calculate ATM biases and vice versa? What is the specific satellite bias the authors are presenting in Figure 12a? In

the caption to Figure 12a, what does it mean to predict biases based on OLCI grain sizes that are a function of ATM grain sizes? How are the OCLI grain sizes a function of the ATM grain sizes?

- Figure 12: Why are these bias curves different from those presented in Figure 4? In Figure 12b a 0.12 m ATM bias corresponds to a grain size larger than 1000 $\mu$m whereas in Figure 4 it occurs at less than 1000 $\mu$m.

- Line 449: The ATM biases in Figure 12b don't really appear to be uniform at around 0.02 m for all grain sizes below 250 $\mu$m as stated in the text. For instance, at 1 $\mu$m the bias is effectively 0. Is there a lower limit the authors would apply to their best feasible correction or do they mean to imply the existence of a constant 0.02 m bias in the ATM data for small grain sizes?

- Line 459: Could the authors elaborate on what they mean by "the robust spread of the distribution" as it is not a familiar metric. Is it similar to the interquartile range or mean absolute deviation? Also, the reason for using this metric as opposed to something like a standard deviation isn't provided until line 471. I recommend including the rationale for choosing this type of deviation metric when it is first introduced.

- Line 471: Could the authors please elaborate why after establishing the use of the robust spread metric, they suddenly switch to using standard deviation? What does presenting both the robust spread and standard deviation add to the manuscript?

- Figure 13c: The y-axis of this plot implies that at least 1000 samples are found in every bias bin. Is this true? Also, I recommend not using 'K' to express one thousand and either write the complete number or label the y-axis as being in thousands.

- Discussion: Throughout the Discussion the authors continually refer to data or results that have been previously presented. To make it easier for the reader I suggest the authors include pointers to the specific figures they are referring to.

- Line 487: Here the authors suggest look angle could be a reason for the larger grain sizes in the wide swath ATM grain size results. What evidence to the authors present that supports this inference? There is no assessment of grain size versus look angle, so it is difficult to assess the validity of this explanation.

- Line 500: The authors state the grain size relationship between the various grain size estimates is not as consistent as they would have hoped for. Could the authors quantify what the consistency is or what they hoped the agreement between the datasets would have been? This sentence is a little jarring because in the sentence right before the authors state the relationships are consistent but then they say the consistency isn't what they were hoping for and that for a substantial portion of data points there is no clear relationship. What is the reader supposed to take away from this?

- Line 574: Here the authors conclude that the ATM grain size estimates are not strongly sensitive to acquisition geometry. This stands in contrast to line 487 where the authors state that larger grain sizes in the wide swath ATM results are due to look angle. Which is it? And again, there is no support for such a statement in the manuscript.

***Technical Comments***

- Line 15: "form" to "from"

- Lines 32-35: This is a very long and meandering sentence that is difficult to follow in its entirety. I would recommend partitioning it to more concise statements. Also please include references to support the specific points the authors are making (e.g., ICESat-2 vs ICESat precision and efficiency, weak absorption of green light by ice).

- Lines 36-40: Again, a long and meandering the sentence. It begins talking about glaciers but then halfway through ice shelves and sea ice is introduced, both time- and space-varying biases are discussed but in different contexts. Please consider partitioning the sentence into more distinct statements.

- Lines 41-49: The authors use terms such as "escaping", "leaving", "returning", and "scattering" all to describe a laser signal that is reflected from the surface back towards the detector. I would recommend choosing one and using it consistently. Also, for a paragraph outlining the fundamental physical processes underlining this study, the lack of references supporting them is surprising. I recommend including references supporting the physical phenomena discussed.

- Line 71: Double use of "waveform"

- Line 79: "return-pulse shape" vs "recorded pulse" vs "waveform" all seem to refer to the same concept, so I would recommend the authors choose one term and use it consistently.

- Line 78-79: Please provide a reference(s) to support this type of direct statement.

- Line 79-83: Please provide a reference(s) supporting the statement on how surface and subsurface effects manifest in different altimetry systems and how easy they can be measured.

- Line 84: Please provide a reference(s) supporting the ATM heritage and evolution.

- Line 91-95: Another winding sentence. I recommend partitioning between the discussion of LVIS and SIMPL.

- Line 146: Please be consistent in the use of "AVIRIS" or "AVIRIS-NG".

- Line 156-157: The authors refer to multiple studies with forward laser waveform models, specifically those making use of diffusion theory to predict the waveform shape, but only one reference is provided. If there is more than one study using a similar model, please include them as well.

- Line 160: Please provide a reference supporting the statement that the diffusion-based approach can produce unphysical results.

- Line 187: Here it is stated that a grain size of 200 $\mu$m is used in the derivation of Figure 1 when the caption states 1000 $\mu$m. Please clarify which is the correct value.

- Line 240: Is the colon in this line is meant to be a period?

- Lines 255-268: There is only one sentence in this paragraph that doesn't begin with either "To" or "For" and that's because it starts with "Figure". I recommend the authors avoid such repetitive writing as it is makes the paragraph difficult to read.

- Line 280: ICESAT-2 to ICESat-2

- Line 289: Please be consistent throughout the manuscript on how Equations are referenced. In this line the authors use parentheses, *(6)*, while in other instances *Eq.*, *Eqn.* and *Equation* have also all been used.

- Line 299: Please be consistent in using ICESat-2 or IS2. In this paragraph and Figure 4 legend, the authors flip back and forth.

- Line 316: The authors have two Section 3's. I imagine this Section and all sub-sections should be renumbered to Section 4.

- Line 365: Here and throughout the manuscript the authors use multiple versions of grain size (grain-size, grainsize, grain size). I recommend the authors use the more common "grain size" consistently throughout the manuscript.

- Line 366: The "For" at the end of the line should not be capitalized.

- Line 370: I believe the authors are referring to Figure 7a here instead of 6a?

- Figure 10: I assume the comma at the end of the caption is meant to be a period?

- Line 452: Please be consistent through the manuscript with how subfigures are referred to. Here the authors use "panel B" but they have also used "XB" such as on Line 425.

- Line 467: I recommend the authors be consistent with their units. Here they use both centimeter and meter units when referring to the same thing.

- Lines 593-598: This sentence is too long. I recommend partitioning it into smaller, more direct statements.

- Line 603-607: This is almost an exact repeat of what is stated earlier in the same paragraph (lines 586-588). Please carefully review the document to check for redundancies.

---

## Author Comment (AC1)

*General Comments*

This manuscript presents a study aimed at predicting biases in ICESat-2 surface elevations due to the subsurface scattering of photons. The authors base their approach on the comparison of airborne laser waveforms acquired with the Airborne Topographic Mapping (ATM) system and grain size estimates from the Airborne Visible/Infrared Imaging Spectrometer – Next Generation (AVIRIS-NG) instrument and the Ocean and Land Colour Instrument (OLCI) on-board Sentinel-3. A subsurface scattering model is employed to derive grain size estimates from the ATM waveforms, which are then compared to the AVIRIS-NG and OLCI observations. Finally, these results are to predict possible biases in ICESat-2 elevations due to estimated grain sizes.

Overall, I found the manuscript very confusing and difficult to read. Many of the comments below will speak more to this point, but there is general a lack of specificity and clarity in how the manuscript is written. I think much of the confusion with regards to the scope of the manuscript comes from how it is initially presented as "Understanding biases in ICESat-2 data…" when there is no actual ICESat-2 data used. With how often ICESat-2 is mentioned in the Abstract and Introduction, the authors build an expectation that at some point what is discussed will be applied back those data, but it never actually happens. Also, as is stated on line 586, to date no biases in ICESat-2 due to subsurface scattering have been observed, which brings into question the overall relevance of the study. Are the authors proposing a solution to a problem that doesn't exist? The broad-scale importance of this work is not clear.

I think the manuscript would benefit from a thorough revision focused on improving context and clarity. I believe the foundational elements (i.e., approach, data analysis, results) of the study are interesting and worthy of publication, but how they have been packaged makes the full arc of the study difficult to follow. I also found the manuscript to be under-referenced. The authors often make direct statements without providing any supporting evidence.

I recommend the authors consider the following points while revising their manuscript. I realize there are a lot of them, but I want to be detailed and clear with regards to where I had trouble following the manuscript to help the authors with their revisions.

> General response: I (Ben) should have spent much more time polishing the first draft of this manuscript before submitting it. A manuscript that is confusing and difficult to read makes being a referee, which is a volunteer job that is uncompensated at best, much harder than it needs to be. I am grateful to referee 1 for all the effort that went into this report, and for the level of detail in the recommended revisions. We have spent a lot of time revising the manuscript to improve the writing, to fix some of the confusing terminology that was in the first version, and have added material to help motivate the methods.
>
> There is no way to deny that the lack of ICESat-2 data in the paper is strange. The reason for it is that the kind of biases we are discussing are not at all straightforward to observe in the field, where measuring even decimeter-scale biases over large areas in remote parts of Greenland is not straightforward. This becomes especially difficult using ATM measurements that suffer the same bias as ICESat-2 (as shown in figure 4, and now noted in the text). Our revised introduction takes more care to motivate the use of biases estimated from ATM waveform measurements as a proxy for ICESat-2 biases, and we have tried to improve the clarity of section 4.6, where we try to put all the pieces together. We hope that the new version of the manuscript is easier to follow, and better makes the points that we hoped to make in the first version.

We respond to the specific comments below, leaving the referee's comments in black, serif font, with our responses in blue, sans-serif. Quotes from the revised study are in italics.

**Specific Comments**

-Abstract: Please provide more context for the last sentence. Why is it important to correct for subsurface scattering biases in ICESat-2 data? Within the context of the overall ICESat-2 height uncertainties, how big of a problem is it if these biases are left unaccounted for?

- We added a sentence to describe the previous modeling results in the context of the ICESat-2 measurement accuracy requirements:
  *As an example, modelling suggests that ICESat-2 might experience a bias difference as large as 0.1-0.2 cm between coarse-grained melting snow and fine-grained wintertime snow (Smith et al., 2018), which is as large or larger than the mission's requirement to measure seasonal height differences to an accuracy better than 0.1 m (Markus et al., 2017).*

-Line 31: Please consider including the corresponding wavelength for ICESat to support the comparison to ICESat-2's 532 nm green laser.

- We now specify that the ICESat laser operated at 1064 nm.

-Line 47: Here weaker scattering is implied as being associated with broader returns and larger delays, but wouldn't weaker scattering be equivalent to stronger absorption in the medium? In the second and last sentence of the paragraph, stronger absorption is associated with tighter photon distributions.

  -We added a clause to the last sentence to explain why more absorption is associated with narrower returns:
  *Light absorption within the scattering medium can also influence the time distribution of returning photons, with stronger absorption producing narrower distributions and smaller net delays because photons are often absorbed by the medium before they can accumulate long delays.*

- Line 50: Someone approaching the manuscript from an ICESat-2 (i.e., individual photon) perspective may be unfamiliar with the concept of a laser "waveform". I'd suggest the authors early in the manuscript define what a waveform is. Here a "return photon timing distribution" provides an excellent opportunity.

  - Added: *(i.e. the measured waveform in an analog lidar, or the distribution of photon timing in a photon-counting lidar).*

-Lines 70-72: The process of using a model for the shape of ATM waveforms to predict grain size and comparing that to airborne and satellite measurements is well laid out. What I continually struggle with is the subsequent extension to ICESat-2. What does it mean to "confirm" that predicted biases in ICESat-2 are the same as those for the ATM case when there are no actual ICESat-2 biases to compare against? Is the authors goal to show 1) that their scattering model is valid when using airborne (ATM and AVIRIS-NG) data, 2) can be applied to Sentinel-3 gain sizes to predict a generic laser bias, and 3) that generic bias compares favorably against ATM data, so that, in principle, one could possibly apply it to ICESat-2? So much of the emphasis in the manuscript is placed on applicability to ICESat-2 but by the end, that link seems like more of an inference/prediction/extrapolation than something that is concretely demonstrated (i.e., confirmed). Furthermore, I suggest the authors be more specific with what they mean by the "sampling" of altimetry measurements and how that affects calibration. What type of "sampling" are the authors referring to?

  • We have revised this paragraph to weaken the claim that we had made about

"confirming", and have removed the statement about sampling:

*In this study, we investigate the scattering properties of Greenland snow and ice surfaces using altimeter waveform shapes, with the goal of developing a correction for the biases that subsurface scattering can introduce into ICESat-2 data. Although this study is motivated by the need to understand biases in ICESat-2 measurements related to subsurface scattering of green light, data from ICESat-2 are rarely suitable for investigation of subsurface scattering biases, because over rough and sloping surfaces, ICESat-2's 11-m footprint leads to a significant random component in the timing of returned photons, which tends to obscure small changes in the timing distribution associated with subsurface scattering. Slope and roughness tend to be largest in low-elevation regions of Greenland* (Nolin and Payne, 2007), *which are the same regions where we expect to see the largest subsurface scattering biases. Instead, we use waveform measurements from the ATM airborne laser-altimetry system to test a previously developed model of subsurface scattering* (Smith et al., 2018) *based on a comparison between the shapes of the returned pulses and pulse shapes expected based on the model. We demonstrate that by adjusting the scattering parameters in the model to match modelled waveforms to measured waveforms we can recover an estimate of the near-surface optical grain size. We test the grain-size estimates recovered from waveform matching by comparing them against grain-size estimates derived from airborne and satellite reflectance measurements. Although this comparison does not suggest a 1:1 linear relationship between waveform-derived grain sizes and reflectance-derived grain sizes, we use a proxy for ICESat-2 biases based on the ATM data to calibrate a correction based on reflectance-derived grain sizes, and demonstrate that the calibrated correction can produce elevation estimates that, averaged over a range of Greenland terrain and surface conditions, are unbiased. The results of this study fall short of a correction that could substantially reduce grain-size-driven biases in ICESat-2 data, and we provide a description of some of the advances in satellite remote sensing that would be needed to more adequately address this problem.*

-Section 2: I know maps are presented in Figure 5, but I would recommend the authors consider including a composite overview map when describing the datasets to help situate the reader.

- Presenting the measurement locations at this stage would involve showing essentially the same figure twice. We now include a note that the measurement locations are shown in section 3.

-Line 96: For those unfamiliar with the ATM data, please provide some specific detail as to what wide-swath and narrow-swath means (e.g., cross-track look angle, width on ground, etc.).

- We now include this information and a table indicating which data were processed from which campaign:

*Waveform measurements in this study come from data collected in Greenland in the 2017 summer campaign, the 2018 spring campaign, and the 2019 spring and summer campaigns. Most of the data used in this study (summarized in table 1) were collected using the ATM narrow-swath scanner, whose 5º full scan angle made measurements over a ~40-m swath on the ground at a flight elevation of 500 m. In 2018 and 2019, the aircraft also carried a wide-swath scanner, whose 30º scan angle produced a ~460-m swath, although we only processed the wide-swath data from the 2019 summer campaign. For both instruments, the laser's incident angle on a flat surface is half the full scan angle, thus 15º for the wide swath and 2.5º for the narrow swath. Data from these campaigns are distributed in the ILNSAW1B and ILATMW1B products* (Studinger, 2018a, b),

-Line 105-108: Here the authors refer to some analysis they have done that underlies the choice

of impulse response function but isn't presented in the manuscript. I would recommend the authors consider including this so the reader can understand why these choices were made.

- • We will include a section in the supplemental material illustrating the problem with the transmitted waveforms digitized during the flights.

- Section 2.2: The authors include a lot of detail regarding the ATM system but almost none for the AVIRIS-NG system. How does this system operate? What does it measure? How big is the field-of-view? How closely did the Basler follow the aircraft with the ATM? I believe these details will help to provide context and clarity for the reader.

- • We have expanded the description the instrument and the campaign: *"To help evaluate whether the ATM-derived waveforms were consistent with the returns we would expect from known surface conditions, we used data collected using AVIRIS-NG (the Airborne Visible/Infrared Imaging Spectrometer, Next-Generation), on a Basler aircraft that followed the aircraft carrying ATM on five subsequent days in the autumn of 2019. AVIRIS-NG measures radiances at 425 different wavelengths between 380 and 2510 nm on a detector array that produces images with 598 across-track samples (Thompson et al., 2018); its ~7.5 km altitude during the 2019 survey produced images on a ~4-5 km-wide swath, with ~6-7 m pixel sizes. These measurements were processed to estimate grain sizes using a technique that uses the strength of an absorption feature in the reflectance spectrum of snow at 1.03 μm as an indicator of snow grain size (Nolin and Dozier, 2000). We rejected one of the data files (the single file collected on 9 September, 2019, and the only file with extensive coverage of sea ice) because while the image appears to resolve a melting surface including a variety of sea-ice features including melt ponds and leads, the range of retrieved grain sizes span a small range ( 90% of values between 164 and 287 μm). The reason why this file should contain anomalous values is not clear, although we note that the sun was lower in the sky than it was for any other file (79º solar elevation, as compared to ~70-72º for other files in the campaign). The remaining 26 data files cover two coast-parallel lines and a few coast-perpendicular lines in northwest Greenland, spanning a range of grain-size conditions from large-grained melting surfaces near the coast to fine-grained surfaces inland, and 17 of these overlapped with available ATM waveform files. Most (~80%) overlapping measurements within these five days were collected within three hours of one another, and to limit how much the surface might have changed between one set of measurements and the other, we compare measurements between the two systems only if the differences between timestamps for the data files are less than 200 minutes. "*

- Line 119: Please provide some example numbers for how much wider the AVIRIS-NG swath is compared to the ATM swaths.

- • We now state that the AVIRIS-NG swath was ~4-5 km wide

- Line 122: The authors refer to AVIRIS-NG data with grain sizes inconsistent with the expected surface conditions that were not included. I would recommend being more specific regarding what evidence exists to suggest that these measurements are erroneous. Otherwise, it could leave the impression that the included data were cherry-picked.

- • We reviewed the files that we had initially excluded and determined that there was really only one that needed to be removed, which was also the only AVIRIS file that measured sea ice. We will revise the text to point to a supplemental figure that includes the comparison between the problematic file and the rest of the data, and revise our results to include the two previously excluded AVIRIS files.

- Line 138: I find this description of the "look-up table" and especially the "time steps"

confusing. Can the authors simplify to something along the lines of "<time-specific> grain sizes are interpolated from the daily maps."? It is also not clear the degree of interpolation the authors are performing using the daily maps. Are the authors interpolating at sub-daily intervals or just over missing time periods due to clouds?

> We agree that we had too much detail here. The paragraph now reads:
> *We compared ATM and AVIRIS grain-size estimates with the OLCI-based estimates by bilinear interpolation into each daily grid; if measurements were marked as invalid in an OCLI map because of the presence of clouds, we derived an estimate based on the previous day's map under the assumption that the grain size had not changed substantially between the two days, and if the previous day's estimate was invalid, we rejected the data point.*

- Line 145: What assumption are the authors referring to here and why does in only apply to "most" of the data used in the study? What about the remainder?

- We have removed this sentence, but to answer the question, if clouds were present, the OLCI measurement was marked as invalid, and if there wasn't a valid measurement from the previous day, we made no comparison.

- Section 3: I think there is something missing from this section, specifically how $\tau^*$ related to $r_{"##}$. $r_{"##}$ does not appear in any of the provided equations so it makes it very hard to follow the development of the waveform model as well as understand Figure 1b.

- We have attempted to clarify this section by modifying equation 3 to include the relationship between $\tau^*$ and $r_{eff}$. Please see my response to the comments on line 186.

- Figure 1a: I am confused by Figure 1a. What is the point of introducing the constant velocity (orange) line? It seems to be to try and communicate the how sensitive $\tau^*$ is to the effective velocity of the medium, but in doing so does it not introduce a physically unrealistic scenario? That is, one where density is fixed as it relates to velocity but still allowed to vary with respect to the optical bulk scattering properties. How do the authors justify ignoring the density sensitivity in one variable in the $\tau^*$ equation, while keeping it in the remainder?

- We discuss this in the text. As you note, the orange line is there to demonstrate that the main sensitivity of T* to density comes about because of the distance between scattering events, not because of the effective velocity. As you observe, the orange line represents an unphysical situation, but since it is there to illustrate a sensitivity of the model, we do not see this as a problem.

- Line 185: Could the authors provide some justification for their choice of 400 kg/m³ as a nominal surface density value, when in-situ measurements across Greenland suggest a nominal surface density of 315 kg/m³ (Fausto et al., 2018; https://doi.org/10.3389/feart.2018.00051)?

- We agree that this choice is arbitrary, and added a statement to that effect to the text, quoting the density values from Fausto et. Al.:
  *Although the choice of 400 kg m⁻³ is somewhat arbitrary, it strikes a balance between the smaller, 270-350 kg m⁻³, densities typical of Greenland snow (Fausto et al., 2018) and the larger, 410-910 kg m⁻³ densities observed in ablation-zone surfaces (Cooper et al., 2018).*

- Line 186: $r_{"##}$ does not appear in Equation 1. Please clarify as to where this is coming from.

- We modified Equation 1 to make clear that the scattering coefficients depend on grain size and density, and the asymmetry factor depends on grain size.

- Equation 3: I recommend the authors provide more clarification on the difference between Equations 3a and 3b. It seems 3a is something akin to an "ideal" model and 3b is how the authors approximate it in this study. If that is the case, is it necessary to have both? What does 3a add that can't be communicated by 3b?

- We have removed equation 3a, leaving only our approximation of the surface return (formerly 3b).

- Equation 3: I recommend providing an equation or a more in-depth discussion for the *G*-term in Equation 3b and how it has been selected. How is the surface roughness parameterized within this term? Is it an analytical or empirical representation of roughness? Has this type of parametrization been used previously? Over what roughness range is the model valid? What does it mean to have surface roughness expressed in time, as is done in Line 214? Without this detailed information, the reader cannot make any inference on whether the specific roughness model included in the model is applicable in this application.

- We expanded on our discussion of how slope and roughness affect the return shape, and now include citations to two studies that treat roughness this way, and include a calculation of how the ATM scan angle affects surface return broadening:

  *We approximate the distribution of photon delays due to slope and surface roughness weighted by the illumination pattern of the laser as a Gaussian function, $G(t, \sigma)$.*
  *Our approximation of the effects of slope and roughness follows studies that modelled satellite laser altimetry waveform shapes (Yi et al., 2005; Smith et al., 2019a). If we assume that the illumination pattern is represented by a two-dimensional Gaussian function with standard deviation $\sigma_b$ illuminated surface is represented well as a rough plane whose normal makes and angle $\varphi$ with the beam direction, and that the roughness produces a Gaussian distribution of elevations relative to the plane with standard deviation $\sigma_r$, then the standard deviation of the Gaussian function, $\sigma$, should be equal to $\frac{2}{c}(\sigma_r^2 + \sigma_b^2 \tan^2(\phi))^{1/2}$. This means that more strongly sloping surfaces should produce broader returns, and that returns from the wide-swath ATM instrument should be broader than those from the narrow-swath instrument. ATM's 0.6-m footprint implies that over a flat surface smooth surface, $\sigma \approx 1$ ns for the wide ($\pm 15$-degree) swath , or 0.1ns for the narrow ($\pm 2.5$-degree) swath.*

- Equation 4: Please provide some explanation for what the *A* and *N* terms in this equation are and what they mean.

  We added:
  *A is a scaling term relating the amplitude of the modelled waveform to that of the measured waveform, and N is the number of samples in the waveform.*

- Lines 257-259: I don't know what this sentence is trying to communicate. I don't understand how the test data have been generated or what "… half the spacing between the searched values." is supposed to mean. I recommend the authors revise this sentence to make it clearer exactly what they have done.

- -To make the paragraph clearer, we revised this passage to read:
  *Our fitting algorithm selects grain sizes based on a set of pre-computed waveforms generated for grain size values separated by 10%, so to demonstrate the worst-case performance of our algorithm, we generated the test data based on grain sizes that were half-way between the grain size values used by the algorithm.*

- Line 270: What is the noise value the authors are referring to here? Please be more explicit in what it is and how it comes about.

- We revised this to specify "the RMS of the noise added to the synthetic waveform."

-Lines 270-278: Much of this paragraph is dedicated to describing the effects ATM amplitude had (past-tense) on the uncertainty in the estimated grain size. The issue I find however, is that the ATM data results have not been covered yet. What do the authors expect the reader to take-away from this paragraph when they have not seen the grain size estimates from the ATM data yet? Furthermore, what does it mean for a surface to be "dark" (Line 275) with respect to laser altimetry? I suggest the authors elaborate or clarify this point. Finally, looking back on this paragraph after reading through the full manuscript, I find it odd that the discussion of precision or uncertainty was not carried through to the actual data analysis. Can the authors quantify the uncertainty in the grain size estimates produced from the analysis of the ATM data that they mention here?

- This part of the study contains techniques for analyzing datasets rather than results, so it needs to come before the reader encounters actual ATM data. We also revised this paragraph to make clear that low-amplitude returns are associated with longer ranges and coarser grain sizes:

  *in areas where fine-grained snow was mixed with coarser-grained surfaces, or where the range to the surface was highly variable because of rugged topography, the strongest returns may have been captured with settings that produced large amplitudes, while the coarser, more distant surfaces had lower amplitude, and thus lower precision grain-size estimates.*

-Line 290: Please consider elaborating further on why negative numbers affect the waveform-median retracker. Is there no other type of retracker that is not sensitive to sensitive to negative numbers that could be used? Also, I would recommend the authors consider including more detail on the two types of retrackers applied to the simulated ICESat-2 data (windowed mean and windowed median) for those who are not familiar with these specific ICESat-2 details. How do they work? The authors are assuming the reader is familiar with this nuanced part of ICESat-2 operations.

- In response to the question about whether a different retracker might be used that is not sensitive to negative numbers: This study is concerned with evaluating biases in the ICESat-2 standard elevation product, so we don't have much choice about which retracker we evaluate.
- We will add material to our introduction explaining that we are specifically evaluating the ATL06 h_li parameter
- We have revised this paragraph to expand on the difficulty in using a median on a waveform that contains negative numbers:
  *Ideally, to evaluate the expected biases in this parameter, we would use measured ATM waveforms to approximate $W_s(t)$, and use the ATM IRF to approximate $W(t)$, which would let us directly use Eq.6 to calculate expected biases with the windowed waveform median as $M(\Box)$. This is not practical, however, because most ATM waveforms include digitizer output that is less than zero (see figure 2). ICESat-2 uses a photon-counting lidar, so the median elevation can be calculated directly from the distribution of photon heights within the window. For a waveform lidar, the waveform median can be approximated under the assumption that the numerical value of the waveform is proportional to the flux of photons into the detector. In the case of ATM, however, the recorded waveforms include negative values, which imply a more complicated relationship between the photon flux and the recorded values, which implies that we cannot calculate the waveform median directly from ATM waveforms. Instead, we model the effects of subsurface scattering on ATL06 biases by using Eq. 3 to generate synthetic scattering-affected waveforms for a range of grain sizes, based on an estimate of the ICESat-2 system IRF.*

•

- Line 293: What is the IRF function for ICESat-2 and where do the authors get it from? I recommend the authors provide more elaboration on this point. Substantial space was given to establishing the ATM IRF in Section 2, while here the ICESat-2 IRF is almost glossed-over.

  - We now specify that this waveform was derived from pre-launch calibration measurements, and provide a citation to Smith et al, 2018.

- Section 3.4: I think a key point I'm struggling to understand is the link between waveform model of Equation 3 and what is measured by a photon-counting laser altimeter such as ICESat-2. It underlies this entire section and the Section beginning on line 452 but it is not clear to me how this works. Perhaps including an example diagram of how the authors extract the more ICESat-2 relevant parameters from the modeled waveforms would help with this?

  - We now provide more detail about how the IS2 median-based elevation is calculated, and how this relates to the way that we calculate the biases using our model. Combined with the improved description of the goals of the study that we provide in our introduction, we hope that this addresses the referee's concerns here.

- Figure 5: I suggest the authors consider including specific dates in the titles of the subfigures instead of the generic "spring" and "summer" labels. The authors refer to earlier and later spring campaigns in the main body (Line 326) and including actual dates in the figure would make this clearer. Also, some of the detailed patterns the authors discuss in the main body (e.g., Line 331) are very difficult to see in Figure 5. I suggest the authors consider including a zoomed in version of the larger maps that highlight exactly what they are talking about.

  - We will modify the titles to specify the months included in each panel. Because the goal of this figure is to provide a broad-scale spatial context for the measurements used in the study, we prefer to maintain a consistent zoom level for all the panels, rather than zooming in as the referee suggests.

- Line 326: When speaking to specific sub-figures, I recommend the authors actually refer to them (i.e., Figure 5b). This helps to make things much clearer for the reader.

  - We have added these references

- Figure 5 and Figure 6a: Please include coordinates for all maps.

  - Including coordinates on maps that are intended to provide illustrations of features in the data (e.g. that two datasets are similar) rather than to illustrate a property of a location (e.g. that the grain size had a particular value at a particular location) can make the layout of the figure more difficult without providing a lot of useful information to the reader. If the study were about Leidy Glacier and its grainsize variability, then the location of the figure would be much more relevant, but in this case, we are simply illustrating that ATM measured grainsize variations. A brief survey of other papers in TC showed that some provide coordinates for all maps, and some do not, and our figures are much easier to generate without coordinates. In the interests of transparency, we are including the corner coordinates for each of our maps in a table in the supplement to the article.

- Figure 6a: The Blues colormap used to represent estimates grain size on top of the Landsat image is very difficult to distinguish. I recommend the authors consider a different colormap that stands out more from the background. Another option would be to segment the grain size estimates based on the ranges presented in Lines 250-251.

  - I (Ben) like the Blues colormap for this figure, in part because the blue-white ramp is analogous to the blue appearance of coarse-grained snow and the white appearance of fine grained snow. To help distinguish the part of the track that

extends to the right of the figure, we revised the figure so that the track is outlined in black.

- Line 350: Here the authors refer to the lower portion of the Leidy glacier as having experienced extensive melting. I recommend the authors include some justification for this as the distance between the larger grain size lower portion and the finer grain size upper portions is not much. Has extensive melting been so highly concentrated in only the lower section?
  - There is a strong elevation gradient across the image, which likely leads to the range of grain sizes. We now specify that our statement about the grain sizes is an interpretation rather than a certainty:
    - *Elevations measured by ATM show that the outlet section of the glacier (near C) is at 400-500 m, and elevation increases to around 1200 m near D. The mapped distribution of grain sizes (panel A) shows little or no subsurface scattering on rock and soil ($r_{eff} \approx 0$), strong subsurface scattering for low-elevation ice ($r_{eff} > 1000\ \mu m$), and weaker subsurface scattering at higher elevations ($r_{eff} < 200\ \mu m$). We suggest that the lower-elevation part of the glacier on the left-hand part of panel 6A has experienced stronger surface melt than the higher-elevation part to the right of panel 6A, which is roughly consistent with the gradient from bluer to whiter tones in the background Landsat image collected two days later.*

- Figure 7a: Please include a colorbar as is done for Figures 9 and 10.
  - We will include a colorbar in the revised manuscript

- Figure 7b: Is there a specific reason as to why the distributions are presented on a log-normal scale? What are the units for the spreads provided in the legend? It seems odd to plot the data on a log-normal scale (especially something like a ratio) and then use the standard deviation. I recommend the authors explain why they expect the ratio between the wide and narrow swath ATM grain sizes to be logarithmically distributed.
  - The caption was incorrect in this case. The spread values quoted are based on the robust spread, which we now define in the methods section.

-Line 363: To me "around" does not reflect the situation presented in Figure 7a. It appears as if the wide swath grain sizes are consistently larger than the those from the narrow swath. Perhaps it would be more representative to use a term like "near"?
  - Revised to 'close to'

- Figure 8: Please provide coordinates for the plot. Also, the authors mainly discuss the upper half of this plot. So would it prudent to zoom into the upper half in order to see the spatial patterns the authors discuss more clearly?
      We chose the bounds of the plot so that the width of the AVRIS swath and the ATM swath could be shown in the same axes. See our comment on figures 5 and 6 for our rationale about coordinates.

- Line 398: Here the authors refer to the comparison of the ATM and AVIRIS-NG gain sizes in Figure 9. I recommend the authors clarify which ATM dataset is being compared. Is it the narrow swath, the wide swath, both?
  - Figure 9 uses only narrow-swath data. We will specify this in the text and in the caption.

-Line 401: Here the authors refer to ATM grain size being equal to or smaller than the AVIRIS-NG grain sizes when the pattern look more like a loss in sensitivity in the AVIRIS-NG results

at small grain sizes (the authors also state this on Line 494). Is this the most appropriate way to describe the results, as being equal to or smaller than? It implies that there is closer agreement between the datasets, whereas it appears more likely to be a numerical artefact.

• We will revise this section to include a brief explanation for the vertical feature:
*This relationship does not hold towards the small-grainsize side of the plot, where the AVIRIS grain sizes are clustered in a near-vertical feature centered around 50 µm. We believe that this is because the AVIRIS algorithm loses some of its sensitivity to grain-size variations around 40-50 µm while, based on our synthetic-data experiments, we expect the ATM retrievals to be sensitive to grain sizes as small as 25 µm*

- Figures 7a, 9, 10, and 11: All these plots show the same type of comparison between grain sizes. As such, I'd recommend the authors consider standardizing them such that the axis limits and colorbars are the same.

• Figures 9, 10, and 11 will have consistent axis limits in the revised MS and will have the same units on the colorbars ("log measurement count, arbitrary units"). It does not make sense to have consistent colorbar limits, because the total number of measurements is not at all the same between the plots. Figure 7a has different axis limits, which allow us to show the points where one ATM sensor or the other converged to the no-scattering waveform.

- Figures 9-11: I recommend the authors elaborate on why different bounds were chosen for the dashed white lines in these plots (i.e., factor of 2 in Figure 9 and factor of 3 in Figures 10 and 11). Also, please ensure the bounds are plotted properly as they look very similar between the factor of 2 and factor of 3 cases.

• These figures will be standardized to use a factor of 3.

- Line 426: Here the authors state that comparison of grain size estimates "… roughly follow the 1:1 line.". Would it be possible to strengthen this argument using a quantitative value such as the correlation coefficient or some other metric?

• The correlation is not especially strong, and there is a substantial background of low-quality OLCI measurements evident in this plot. This general statement of the relationship between the two datasets is about as strong a statement as the data allow.

- Lines 442-444: I find this sentence very confusing, and I don't know what the authors are trying to communicate. Why is it important to use OLCI grain sizes to calculate ATM biases and vice versa? What is the specific satellite bias the authors are presenting in Figure 12a? In the caption to Figure 12a, what does it mean to predict biases based on OLCI grain sizes that are a function of ATM grain sizes? How are the OCLI grain sizes a function of the ATM grain sizes?

• As written, this section was confusing. We are assuming that biases calculated from ATM grain sizes are a good proxy for ICESat-2 biases, because even if ATM waveform shapes can't be precisely translated into grain sizes, the way in which subsurface scattering affects the ATM waveform should be the same as it affects the ICESat-2 waveform. The question then is how well we can use the independent grain-size data to predict ICESat-2 biases. In the original paper, we made this confusing concept even more confusing by referring to OLCI-derived biases as "satellite-derived biases, " and since ICESat-2 and OLCI are both (roughly speaking) satellites, the results were not easy to follow. We have improved on this by specifying which satellite we are talking about throughout the paragraph, by introducing the section more gently, and by introducing the rationale for this comparison in our methods section.

- Figure 12: Why are these bias curves different from those presented in Figure 4? In Figure 12b a 0.12 m ATM bias corresponds to a grain size larger than 1000 $\mu$m whereas in Figure 4 it occurs at less than 1000 $\mu$m.

- This was a case of confusing writing on our part. The bias plotted in figure 12 is the ICESat-2 bias predicted from the ATM grain size, so it is equivalent to the orange curve in figure 4. Comparing those two curves, each has a value of around 0.08 m at 1000 μm. This section will undergo a careful rewrite to make sure it is clear what we are comparing and why.

Line 449: The ATM biases in Figure 12b don't really appear to be uniform at around 0.02 m for all grain sizes below 250 $\mu$m as stated in the text. For instance, at 1 $\mu$m the bias is effectively 0. Is there a lower limit the authors would apply to their best feasible correction or do they mean to imply the existence of a constant 0.02 m bias in the ATM data for small grain sizes?

- The problem is with the OLCI grain sizes, not the ATM grain sizes. We observe that OLCI becomes inconsistent with ATM for small grain sizes, and use the tuning exercise to determine a cutoff grain size below which we will not trust OLCI, and to determine a constant bias value to use for OLCI grain sizes below this cutoff.

- Line 459: Could the authors elaborate on what they mean by "the robust spread of the distribution" as it is not a familiar metric. Is it similar to the interquartile range or mean absolute deviation? Also, the reason for using this metric as opposed to something like a standard deviation isn't provided until line 471. I recommend including the rationale for choosing this type of deviation metric when it is first introduced.

- We now include "robust spread" in our methods section.

- Line 471: Could the authors please elaborate why after establishing the use of the robust spread metric, they suddenly switch to using standard deviation? What does presenting both the robust spread and standard deviation add to the manuscript?

- We explain as follows:

  *Performing this analysis with robust statistics (i.e. the median and robust spread) shows how the correction works for typical locations on the ice sheet, which we would expect to fall in the middle of our distribution of residuals, because the robust statistics are less sensitive than their standard counterparts, the mean and standard deviation. Because many users of altimetry data will explicitly or implicitly perform their analysis using non-robust statistics, we repeat the analysis using the mean and the standard deviation of the corrected datasets. This yields similar optimum $B_0$ and $r_{thr}$ values (0.014 m and 250 μm, respectively) for the zero-mean-residual model with the smallest standard deviation, but finds that for this model, the standard deviation is approximately the same as that for the non-optimized correction (0.014 m in either case). This shows that with the right parameters, the correction can produce a near-zero corrected mean, but cannot make a substantial improvement in the standard deviation of the corrected data.*

- Figure 13c: The y-axis of this plot implies that at least 1000 samples are found in every bias bin. Is this true? Also, I recommend not using 'K' to express one thousand and either write the complete number or label the y-axis as being in thousands.

- This is the correct interpretation. We will revise this figure to indicate the thousands in the caption.

- Discussion: Throughout the Discussion the authors continually refer to data or results that have been previously presented. To make it easier for the reader I suggest the authors include pointers to the specific figures they are referring to.

- We agree that this was confusing and are adding references to each section and figure throughout the discussion.

- Line 487: Here the authors suggest look angle could be a reason for the larger grain sizes in the wide swath ATM grain size results. What evidence to the authors present that supports this inference? There is no assessment of grain size versus look angle, so it is difficult to assess the validity of this explanation.

- We now provide a more detailed explanation, and refer to our more thorough treatment of return pulse shape in section 3.2:

  *The comparison of measurements between the narrow and wide-swath instruments (fig. 7) shows that ATM-based estimates of grain size are consistent to within a factor of two or better between two independent instruments, and are not strongly influenced by measurement geometry except at small grain size, where the larger angle between the wide-swath beam and the surface produces blurring of the returned waveform. Based on our modelling results (fig. 3) and the expected relationship between incident angle and return pulse width (section 3.2), we expect this to result in larger scatter and bias in the wide-swath grain-size estimates. As estimates of grain size, the two sets of measurements have biases and uncertainties due to our assumptions about the density of the snow, but as measurements of photon delays due to subsurface scattering, they are reasonably consistent and should be useful in predicting biases in ICESat-2 data*

- Line 500: The authors state the grain size relationship between the various grain size estimates is not as consistent as they would have hoped for. Could the authors quantify what the consistency is or what they hoped the agreement between the datasets would have been? This sentence is a little jarring because in the sentence right before the authors state the relationships are consistent but then they say the consistency isn't what they were hoping for and that for a substantial portion of data points there is no clear relationship. What is the reader supposed to take away from this?

- We have weakened our first statement about the agreement between the datasets (now "broad agreement"). The rest of this seems like it says what we want to say— the agreement between the datasets is imperfect, unlike the point-for-point agreement that we might have hoped for. The rest of the paragraph explains how this disagreement came about. I hope it is less jarring without the repetition of the word "consistent."

- Line 574: Here the authors conclude that the ATM grain size estimates are not strongly sensitive to acquisition geometry. This stands in contrast to line 487 where the authors state that larger grain sizes in the wide swath ATM results are due to look angle. Which is it? And again, there is no support for such a statement in the manuscript.

- It's in the comparison between the wide-swath and the narrow-swath systems that we show that the grain-size recovery isn't very sensitive to the measurement geometry. One set of measurements is made at a 2.5 degree incidence angle, and the other is made at a 15-degree incidence angle, and except at the smallest grain size, the two agree. We revised the paragraph to clarify:

  *We showed that measurements are consistent between two independent versions of the same instrument flown on the same aircraft at the same time with different look angles, showing that the grain-size recovery is repeatable, and is not strongly sensitive to the geometry of the measurements, except at small grain sizes for which the larger incident*

*angles associated with the wide-swath scanner begin to degrade the sensitivity of the system*

**Technical Comments**

- Line 15: "form" to "from"
  - • Fixed

- Lines 32-35: This is a very long and meandering sentence that is difficult to follow in its entirety. I would recommend partitioning it to more concise statements. Also please include references to support the specific points the authors are making (e.g., ICESat-2 vs ICESat precision and efficiency, weak absorption of green light by ice).
  - • We have now split this sentence into three and added references.

- Lines 36-40: Again, a long and meandering the sentence. It begins talking about glaciers but then halfway through ice shelves and sea ice is introduced, both time- and space-varying biases are discussed but in different contexts. Please consider partitioning the sentence into more distinct statements.

  - • We simplified this sentence as follows: *These biases have the potential to interfere with ICESat-2's primary mission goals of precisely measuring elevation changes over glaciers, ice sheets, and ice shelves (Markus et al., 2017) because time varying biases in ICESat-2 measurements could produce spurious signals that might be interpreted as ice-sheet mass changes. Likewise, spatially varying biases in ICESat-2 measurements over sea ice might be interpreted as variability in freeboard and thus ice thickness.*

- Lines 41-49: The authors use terms such as "escaping", "leaving", "returning", and "scattering" all to describe a laser signal that is reflected from the surface back towards the detector. I would recommend choosing one and using it consistently. Also, for a paragraph outlining the fundamental physical processes underlining this study, the lack of references supporting them is surprising. I recommend including references supporting the physical phenomena discussed.
  - • We chose "escaping" as the preferred term, and added a description of a hypothetical lidar system that measures a delayed, broadened return: *The problem of biases in altimetry data that result from subsurface multiple scattering in snow and ice has been described in previous studies (Harding et al., 2011; Smith et al., 2018). Light is reflected from snow surfaces primarily by multiple scattering, where each photon scatters off many snow grains before escaping the snowpack (Wiscombe and Warren, 1980; Warren, 1982). When light scatters from granular materials that absorb light strongly, only those photons that have scattered a small number of times escape the surface. By contrast, light scattering from weakly absorbing granular materials may enter the surface and scatter from tens or hundreds of grains before escaping again. The extra distance travelled during these subsurface scattering events delays the return of the photons to the surface, so light escaping the surface includes photons that have travelled a distribution of long and short paths. A lidar system measuring the range to a weakly absorbing surface will measure returning photons that have a longer mean travel time and a broader distribution of return times than it would from a non-scattering or strongly absorbing surface. The mean delay of the photons and the shape of the returning pulse depend on the scattering properties of the material, with lower densities and*

*coarser grain sizes corresponding to weaker scattering, broader returns, and longer delay times. Light absorption within the scattering medium can also influence time distribution of returning photons, with stronger absorption producing narrower distributions and smaller net delays because photons are often absorbed by the medium before they can accumulate long delays. The distribution in time of reflected energy thus offers the potential to provide information about the optical properties of snow and ice surfaces.*

- Line 71: Double use of "waveform"
    - • Fixed
- Line 79: "return-pulse shape" vs "recorded pulse" vs "waveform" all seem to refer to the same concept, so I would recommend the authors choose one term and use it consistently.
    - • We have chosen "waveform" as the preferred term.

- Line 78-79: Please provide a reference(s) to support this type of direct statement.
    - • We now state that this is *our* model of how measured waveforms relate to instrument and surface parameters.

- Line 79-83: Please provide a reference(s) supporting the statement on how surface and subsurface effects manifest in different altimetry systems and how easy they can be measured.
    - • We intended this paragraph to provide a survey of the different systems that are available, to help explain why we chose ATM. These systems have not previously been evaluated for subsurface scattering, so there is no citation available. To avoid giving the impression that we are withholding information, we now state that this is our opinion.
- Line 84: Please provide a reference(s) supporting the ATM heritage and evolution.
    - • We now cite two papers that include descriptions of different stages in ATM evolution

- Line 91-95: Another winding sentence. I recommend partitioning between the discussion of LVIS and SIMPL.
    - • Now broken in two

- Line 146: Please be consistent in the use of "AVIRIS" or "AVIRIS-NG".
    - • We now use 'AVIRIS-NG' throughout
- Line 156-157: The authors refer to multiple studies with forward laser waveform models, specifically those making use of diffusion theory to predict the waveform shape, but only one reference is provided. If there is more than one study using a similar model, please include them as well.
    - • We now provide three references.

- Line 160: Please provide a reference supporting the statement that the diffusion-based approach can produce unphysical results.
    - • Reference added.

- Line 187: Here it is stated that a grain size of 200 $\mu$m is used in the derivation of Figure 1 when the caption states 1000 $\mu$m. Please clarify which is the correct value.
    - • This was a mistake in the first version. 200 µm is correct, and we have updated the caption.

- Line 240: Is the colon in this line is meant to be a period?
    - • Changed to a period.

- Lines 255-268: There is only one sentence in this paragraph that doesn't begin with either "To" or "For" and that's because it starts with "Figure". I recommend the authors avoid such repetitive writing as it is makes the paragraph difficult to read.

- Revised:
  - *To evaluate the resolution and accuracy of this fitting procedure, we generated a set of test waveforms based on $I_{est}(t)$, for a range of grain sizes, pulse amplitudes, and broadening values. We assessed the sampling distribution of the recovered grain-size estimates by generating 256 different waveforms for each combination of parameters, adding random (Gaussian-distribution) values with a standard deviation of two digitizer counts to each sample, and applying our fitting algorithm to each. Our fitting algorithm selects grain sizes based on a set of pre-computed waveforms generated for grain size values separated by 10%, so to demonstrate the worst-case performance of our algorithm, we generated the test data based on grain sizes that were half-way between the grain size values used by the algorithm. Figure 3 shows the relationship between the specified and recovered grain size for small amplitudes and a range of broadening values (A = 90, $\sigma$ =0, 1, and 2 ns), and for large amplitudes and small broadening values (A=225, $\sigma$ =0 ns). For the high-amplitude waveforms with little broadening (A=255, $\sigma$ =0 ns), the fitting procedure consistently recovers grain sizes as small as 20 $\mu m$, converging to either the next larger or the next smaller grain size value among the searched values (separated by 10%) with a moderate preference for the next smaller value, giving recovered values whose distribution width ($5^{th}$ to $95^{th}$ percentile) is on the order of 10%. At smaller amplitudes and larger roughness values, the width of the recovered distribution increases with decreasing grain size, with distributions spanning around a factor of 5 for $r_{eff}$=50 μm and $\sigma$ =2 ns. For the smallest input grain sizes and the small-amplitude rough input, the waveform that best fit the simulated waveform was often the one with no scattering, so the bottom of the distribution is not constrained on a log scale.*

- Line 280: ICESAT-2 to ICESat-2
  - Fixed

- Line 289: Please be consistent throughout the manuscript on how Equations are referenced. In this line the authors use parentheses, *(6)*, while in other instances *Eq.*, *Eqn.* and *Equation* have also all been used.

  - We now use 'Eq.' throughout.

- Line 299: Please be consistent in using ICESat-2 or IS2. In this paragraph and Figure 4 legend, the authors flip back and forth.

  - We replaced all instances of IS2 with ICESat-2

- Line 316: The authors have two Section 3's. I imagine this Section and all sub-sections should be renumbered to Section 4.
  - Fixed

- Line 365: Here and throughout the manuscript the authors use multiple versions of grain size (grain-size, grainsize, grain size). I recommend the authors use the more common "grain size" consistently throughout the manuscript.
  - We now use "grain size" for the noun form, and "grain-size" for the compound modifier.

- Line 366: The "For" at the end of the line should not be capitalized.
  - Fixed

- Line 370: I believe the authors are referring to Figure 7a here instead of 6a?
  - Fixed

- Figure 10: I assume the comma at the end of the caption is meant to be a period?
  - Fixed

- Line 452: Please be consistent through the manuscript with how subfigures are referred to. Here the authors use "panel B" but they have also used "XB" such as on Line 425.
  - We have revised the manuscript to use XB

- Line 467: I recommend the authors be consistent with their units. Here they use both centimeter and meter units when referring to the same thing.
  - Revised to use meters for both.

- Lines 593-598: This sentence is too long. I recommend partitioning it into smaller, more direct statements.
  - Revised:
    *Improvements in satellite-derived and model-derived estimates (Mei et al., 2021; Painter et al., 2009) of grain size are a potential way to improve the precision of a correction of this kind. One avenue for improvement might be to derive grain-size estimates from satellites with resolution finer than the half-kilometer OCLI data used here. A similar correction using LANDSAT and/or Sentinel-2 data could provide data at 30-meter resolution, although with coarser time resolution and with a less optimal selection of spectral bands.*

- Line 603-607: This is almost an exact repeat of what is stated earlier in the same paragraph (lines 586-588). Please carefully review the document to check for redundancies.
  - We removed the restatement at lines 603-607.

---

## Author Comment (AC2)

**General comments**

This manuscript provides a thorough description about retrieving snow grain sizes from airborne LiDAR and investigating their impact on penetration biases in ICESat-2 elevation data. It is very impressive to see the recovery of new information from datasets that were collected in the past (that were not necessarily customized to retrieve this information). The data and methods are dense but described in detail. The authors make this research immediately applicable by linking recovered snow grain sizes with elevation bias corrections for ICESat-2.

Most of my concerns can be addressed by improving the writing style (see specific comments). My major gripe is the use of "Figure X shows" at the start of paragraphs. It's OK once or twice but it becomes tiresome when every paragraph of the results starts with these words. I recommend that the authors revise some of the first sentences of these paragraphs.

I was also a little underwhelmed by the correlations between ATM/AVIRIS grain sizes with satellite-derived grain sizes. But the authors provide some ideas for the differences which I think this is sufficient for the current scope of the paper.

**Response to general comments**

> Our first submitted version of the manuscript was much rougher than it should have been. Although referee 2 seems to have followed the logic of the study, it can't have been easy, and my (Ben's) writing style was subpar. We thank referee 2 for the kind words, and for the detailed recommendations for revisions. Being a referee is hard and often thankless work, all the more so when the study being reviewed is not presented clearly.

> We have done a lot of rewriting in response to the two reviews and made an effort to strengthen the topic sentences in our results section to avoid the "Figure X shows" problem mentioned above.

> Like Referee 2, we were also underwhelmed by the comparisons between OLCI and ATM/AVIRIS grain sizes. It seems that measuring grain size from space is not easy, and we hope that better techniques and data might help in the future (as we discuss). Even so, we do show that the OLCI-based correction improves the biases in the expected biases, as calculated from the ATM waveforms. Like many other studies, this one will not necessarily solve the problem it sets out to solve, but it should point in the direction of a solution that future studies may achieve.

**Specific comments**

Our responses are interpolated below, in blue.  All quotes from the revised manuscript will be in italics.

L33: There are a lot of "efficients" in the first paragraph. It would be useful to clarify why these detectors are so "efficient". Are they sensitive? Low SNR? Energy efficient?

- We now specify that the detectors are "highly sensitive."

L32-25: Long, wordy sentence, consider splitting.

- Done:
  *These biases are relevant to interpretations of ICESat-2 altimetry measurements over glaciers because ICESat-2 was designed to make precise measurements of glacier elevation change, and time varying biases in ICESat-2 measurements over glaciers and ice shelves can produce spurious signals that might be interpreted as ice-sheet mass changes.  Likewise,  spatially varying biases in measurements over sea ice might be interpreted as variability in freeboard and thus ice thickness (Harding et al., 2011; Smith et al., 2018).*

L36: Just glaciers? Or ice sheets as well? It seems like these two terms are being used interchangeably which is at odds with the first couple of sentences.

- Added ice sheets.

L41-49: Even though this is a well-known phenomenon, it might be useful to add some references here which describe this in more detail.

- We added citations to the previous papers treating this phenomenon (Harding et al, 2011, and Smith et al, 2018).

L56-59: I would need access to these manuscripts to judge overlap and novelty of this paper.

- Both papers are close to publication, and we will provide the drafts as assets in our resubmission of this manuscript

L84: Define acronym on first use of term "ATM" (L75) rather than here.

- In fact, the acronym is defined in the abstract (L15 or thereabout). We will delete the definition here

L88: Diameter?

- Yes. Now specified.

L103: Instruments? Surely there is only one LiDAR or were LVIS and SIMPL also onboard? It may be useful to name the lidar sensor given that ATM is defined as a suite of instruments (or revise L84).

- We now specify "ATM instruments".

L110-112: Might be useful to clarify the difference in swath widths here or L96 when it is first mentioned. Or are these two different sensors? Either way I think some general tightening of terminology is needed in this section.

- This is now covered by a more extensive description of the two systems early in the ATM section.

L114: "Verify" seems a bit strong. Validate or evaluate might be better.

- We will replace "verify" with "help evaluate whether"

L123: This raises the question about how many data files were excluded from the analysis. Are all 26 data files from the same five-day period in 2019 when ATM was followed by AVIRIS-NG?

- We reviewed the files that we had initially excluded and determined that there was really only one that needed to be removed. We have revised the text to point to a supplemental figure that includes the comparison between the problematic file and the rest of the data.

L134: Consider revising because the way it's written makes it sound like Gallet et al. (2009) validated snow-grain sizes from OLCI which was launched in 2016.

- We deleted the reference to Gallet et al. That reference is contained in the Vandecrux paper, which should be adequate documentation of the comparison.

L143: What was the threshold for removing data points that have not been recently updated?

- We do not compare points separated by more than a day (now stated in the text here)

L213: It would be useful to briefly remind readers how the IRF was measured here.

- Added: *"measured using a calibration target with no significant subsurface scattering on 9 March 2018"*

L280: Is the satellite not named "ICES**at**-2"?

Fixed.

Fig. 5: AVIRIS is misspelled in Panel E

Fixed

L361: It would be useful to name the two ATM sensors since this is the first sentence of the paragraph

Done

L398: Please could you clarify if this is **all** coincident grain sizes from Summer 2019 or just a sample.

We now specify that this is all available ice-sheet ATM data, but excludes a single AVIRIS-NG transect on sea ice.

L401: "comes about" is clumsy, consider revising.

Replaced with "is."

Fig. 9: What is the justification for the position of the dashed lines? It would be useful to clarify that here.

- We specify this in the caption: *To help illustrate the magnitude of the difference between the datasets, we plot two dashed lines that show the ATM : 3 x AVIRIS-NG (upper) and ATM : 1/3 x AVIRIS-NG (lower) relationships*

Fig. 10: Same comment about the dashed lines as above.

Also clarified.

L411: Which years?

- All of our AVIRIS data are from the 2019 summer survey, which we now specify:
  *"Figure 10 shows a comparison between AVIRIS-NG-derived grain sizes from the summer-2019 survey and OLCI-derived grain sizes"*

Fig. 11: I think the satellite grain size should be on the y-axis to be consistent with Fig. 10.

We have tried to make our figures so that the less familiar dataset is plotted on the y axis. Since ATM grainsize is the newer, more experimental dataset, we plotted figure 11 that way.

Fig. 12: It took me a minute to figure out what this figure was showing and I think it is because the word "satellite" is being used to refer to both Sentinel-3 grain size and ICESat-2 range bias. Please modify the axes labels to clarify. Also if the black lines show the modeled range bias as a function of grain size then the x-axis label should just be "grain size" and the legend should provide information about where the grain sizes came from.

L440-449: Be specific about which satellite.

We now specify OLCI

L485-489: I think the first half of this paragraph should be removed (or placed later in the discussion) because, as it is written, it seems like the main takeaway is the consistency of the different snow grain size estimates. However, there are substantial biases between the estimates (Figs. 9-11) which the authors are up front about later in the discussion and should be the focus.

L532: Again, which satellite measurements?

We now use "OLCI" instead of satellite

L542-543: This statement seems at odds with L584-585 which states that elevation biases could be decimeter scale. Consider revising.

This may be a misunderstanding. At 542-43, we specify here that there is no problem in the small-grain size regime. The large biases to which we refer in 584-585 are in areas with large grain sizes.

**Citation**: https://doi.org/10.5194/tc-2023-147-RC2

---

## Referee Report (RR1)

**tc-2023-147**

**General Comments**

As Reviewer 1 of the initial submission, I want to start off by sincerely thanking the authors for their efforts in revising their manuscript in response to the comments they received. Overall, I found this updated version of the manuscript much clearer and more straightforward to follow. Most of the comments I have on this updated version are technical issues with only a few minor comments related to specific points. Once addressed I believe the manuscript would be ready for publication within TC.

**Specific Comments**

Lines 249-252: Would one expect σ, and therefore the illumination pattern contribution in Equation 3, to evolve through the ATM swath as the incidence angle changes? The manuscript seems to imply that σ is a constant for the wide (1.0 ns) and narrow (0.2 ns) swath. But wouldn't σ be the same for overlapping wide and narrow swath incidence angles (i.e., in the 2.5° incidence angles around nadir)? I am also missing the connection between the quantified σ values (1.0 and 0.2 ns) and the 0.7 m ATM footprint. Could the authors elaborate a bit more on how they arrive at these values?

Line 485: Here the authors suggest the larger incidence angle as a possible reason for the larger wide-swath grain sizes in Figure 7. Does Figure 7 not represent a point-to-point comparison of the narrow and wide swath grain sizes? If so, I would have expected the only points where this type of comparison is possible to all lie within the overlapping strip near nadir and where the wide and narrow swath incidence angles are equal. Is there something I am missing with how the authors are comparing the two ATM datasets? How are the authors comparing grain sizes from the extreme ends of their wide swath dataset (i.e., the largest incidence angle) with the narrow swath data that don't extend to the same crosstrack extent?

Section 4.5: I still have trouble following exactly what is happening in this section. For example, on Line 564 where the authors write "… the ICESat-2 bias predicted based on OLCI measurements as a function of ATM-derived grain size.". To me, this reads as though the authors are using ATM grain sizes to calculate OLCI grain sizes to calculate ICESat-2 biases, which I have trouble following the logic behind. Maybe it is the use of "… as a function of …" that is causing the confusion and could "… compared to …" be used to equivalent effect? I would also suggest the authors consider revising the y-axis labels in Figure 12 as, as far as I understand, it is not OLCI or ATM biases they are plotting but modeled ICESat-2 biases based on OLCI and ATM grain sizes (i.e., for Figure 12b, essentially combining and turning the y-axes from Figure 11 into a range bias). I think this may also help clarify things.

**Technical Comments**

Line 44: doubled Harding et al. (2011) citation

Line 68: missing space after the Fair et al. (2024) citation

Line 102: "ATM (the Airborne Topographic Mapper) makes laser-altimetry…"

Equation 1: I don't think $r_{eff}$ is ever explicitly defined in the text

Line 246: "… whose normal makes an**d** angle …"

Lines 247 and 249: φ versus ϕ. I would recommend standardizing the notation

Equation 4: the $r_{eff}$ notation used up to this point seems to have been replaced with $r_o$

Line 329: do the authors mean when the SNR is *low*? It appears they are pointing to the upper right portion of Figure 3 and the similar error bars between the $P_{max}$=225 (high SNR) and $P_{max}$=90 (low SNR) scenarios.

Line 355: here $h\_li$ notation is used whereas on line 338 it is $h_{li}$. The $h\_li$ notation appears at other points in the manuscript as well (e.g., Line 388).

Line 361: Is "Over" meant to be capitalized?

Line 403: "… values of $B_0$ and $r_{thr}$ **that** minimize …"

Line 418: Scambos et al. (2012) does not appear in the reference list

Line 423: "… (Fig. 5**c**) …"

Figures 9 and 10: I thank the authors for homogenizing AVIRIS and AVIRIS-NG in their revised manuscript, but I would suggest also carrying this through the x-axis labels in these two figures.

Line 545: "… (**F**ig. 11a) …"

Line 561: "… (Fig. **4**) …"

Line 565: should there be a unit give to $10^{0.25}$? Perhaps $\mu m$?

Figure 13: In the caption, two different fonts and font sizes are used when referring to Equations on Line 611 and Line 615.

Line 632: Missing space prior to the Fair et al. (2024) citation.

Lines 735-736: "… satellite-driven grain-size estimates of providing estimates that would …" could the phrasing here be improved?

---

## Author Response (AR2)

**Authors' response, second round of reviews.**

We thank Dr. Sørensen for taking this manuscript through a second round of reviews. The referees have continued to offer helpful comments, and we attach our responses to their reports below. In addition to the responses to the referees' comments, we have also added an *author contributions* section and have revised the caption to figure 6 to respond to comments from the validation of the previous draft.

All the best
-The authors.

**Referee 1 report, second round**

Thanks again to Referee 1 for their attention to our manuscript. We have clarified the text that led to a misunderstanding about incidence angles for the ATM lasers (described below, comments on lines 249-252 and 485) and have tried to improve the structure of the paragraphs introducing the evaluation of OLCI-based corrections for subsurface scattering. We followed almost all of the referee's suggestions for revisions, which are described inline with the review below. Our responses are in blue, sans serif, and quotes from the revised manuscript are in *italics*.

tc-2023-147
General Comments
As Reviewer 1 of the initial submission, I want to start off by sincerely thanking the authors for their efforts in revising their manuscript in response to the comments they received. Overall, I found this updated version of the manuscript much clearer and more straightforward to follow. Most of the comments I have on this updated version are technical issues with only a few minor comments related to specific points. Once addressed I believe the manuscript would be ready for publication within TC.

Lines 249-252: Would one expect s, and therefore the illumination pattern contribution in Equation 3, to evolve through the ATM swath as the incidence angle changes? The manuscript seems to imply that s is a constant for the wide (1.0 ns) and narrow (0.2 ns) swath. But wouldn't this be the same for overlapping wide and narrow swath incidence angles (i.e., in the 2.5° incidence angles around nadir)? I am also missing the connection between the quantified values (1.0 and 0.2 ns) and the 0.7 m ATM footprint. Could the authors elaborate a bit more on how they arrive at these values?
Line 485: Here the authors suggest the larger incidence angle as a possible reason for the larger wide-swath grain sizes in Figure 7. Does Figure 7 not represent a point-to-point comparison of the narrow and wide swath grain sizes? If so, I would have expected the only points where this type of comparison is possible to all lie within the overlapping strip near nadir and where the wide and narrow swath incidence angles are equal. Is there something I am missing with how the authors are comparing the two ATM datasets? How are the authors comparing grain sizes from

the extreme ends of their wide swath dataset (i.e., the largest incidence angle) with the narrow swath data that don't extend to the same crosstrack extent?

Response: We are sorry for the reviewer's confusion here. When we introduced ATM, we mentioned that it uses "a conically scanning lase," but we did not spell out what that means for the geometry of the beam with the surface. A conical scanner always has the same off-nadir angle (at least for an aircraft in level flight) so it doesn't make any difference whether the points are collected along the aircraft flightline or off to the side- the incidence angle is approximately the same, give or take the surface slope. The point-to-point correspondence in figure 7 does not imply that the aircraft was in exactly the same position for the measurements being compared, just that the measurements were in the same position.
We have updated the text to make this point clear:
*"Note that because ATM uses a conical scanning mechanism, each scanner's beam will intersect a flat surface with an incidence angle equal to the scanner's off-nadir angle. "*

Section 4.5: I still have trouble following exactly what is happening in this section. For example, on Line 564 where the authors write "… the ICESat-2 bias predicted based on OLCI measurements as a function of ATM-derived grain size.". To me, this reads as though the authors are using ATM grain sizes to calculate OLCI grain sizes to calculate ICESat-2 biases, which I have trouble following the logic behind. Maybe it is the use of "… as a function of …" that is causing the confusion and could "… compared to …" be used to equivalent effect? I would also suggest the authors consider revising the y-axis labels in Figure 12 as, as far as I understand, it is not OLCI or ATM biases they are plotting but modeled ICESat-2 biases based on OLCI and ATM grain sizes (i.e., for Figure 12b, essentially combining and turning the yaxes from Figure 11 into a range bias). I think this may also help clarify things.

We agree that this section is difficult and have tried to make the description of Fig. 12 easier to follow by separating the description of the plot from the description of the results it conveys, and by clarifying our terminology about what biases are derived from what sensor. We have also changed the axis y labels to: "ICESat-2 bias predicted from OCLI" and "ICESat-2 bias predicted from ATM." We did not change the orientation of the plots, because the derived quantity in these plots is the estimated range bias, calculated for different values of ATM or OLCI grain size.

Section 4.6 now reads:
*Comparing grain sizes estimated from the different sensors (Figs. 9-11) demonstrates the consistency (or lack thereof) between the datasets, but to address the usefulness of OLCI data in correcting biases in ICESat-2 data, we need to compare biases predicted for ICESat-2 based on OLCI with biases estimated based on ATM waveforms. In these comparisons, the accuracy of the sensor is most important for large grain sizes because ICESat-2 biases predicted by our model (Fig. 4) are approximately zero for small grain sizes, so any correction we calculate will be small, with larger corrections expected for larger grain sizes.*

*If we assume that the ICESat-2 range biases predicted from the ATM waveforms are approximately correct, we can estimate the accuracy of OLCI-derived predictions of ICESat-2 biases in two ways: We can calculate the distribution of OLCI-derived predictions of ICESat-2 range bias for groups of ATM grain-size estimates (Fig. 12a), and we can calculate the distribution of ATM-derived predictions of ICESat-2 range bias for groups of OLCI-derived grain-size estimates (Fig. 12b). In Fig. 12a, we collect groups of ATM grain-size estimates in logarithmic bins with a spacing of $10^{0.25}$ μm and calculate the median and robust spread of biases of the ICESat-2 biases predicted from the corresponding OCLI grain sizes. In Fig. 12b, we reverse this sampling and calculate the distribution of ICESat-2 biases predicted from ATM measurements for groups of OCLI-estimated grain sizes. In each set of axes, we plot the modeled relationship between grain size and range bias for reference.*

*The two plots in Fig. 12 cover different ranges of grain sizes because of the different ways that the two sensors sample the ice sheet. Fig. 12a includes large values of grain size from ATM (up to around 11000 μm) because single ATM measurements occasionally sample features on the surface with large grain sizes and includes no ATM measurements with grain sizes smaller than 30 μm because for smaller grain sizes, ATM often reports zero scattering. In Fig. 12b, grain sizes larger than 2000 μm do not appear, because the 1-km OLCI pixels rarely measure the small features where coarse grain sizes are observed. For the smallest OLCI-derived grain sizes, it appears that ATM often returned no-scattering estimates, so the estimated bias is effectively zero for both datasets.*

*On a per-ATM-waveform basis (Fig. 12a), OLCI bias estimates underestimate the sensitivity of ICESat-2 biases to grain size, especially for large ATM-derived grain sizes. This is likely because OLCI does not resolve small-scale coarse-grained features that are resolved by ATM (e.g. Fig. 8). In Fig. 12b, where the data are binned based on OLCI-derived grain size we see a closer match between the ICESat-2 biases predicted based on the ATM data and those predicted based on the OLCI measurements, at least for OLCI-estimated grain sizes larger than around 250 μm. At smaller grain sizes, the ATM-derived ICESat-2 bias estimates deviate from the OLCI biases, with a roughly uniform value close to 0.02 m for OLCI-derived grain sizes between 20 and 100 μm, a small peak for OLCI biases close to 15 μm, and approximately zero bias for finer grain sizes. This better correspondence shows that when OLCI-derived grain-size estimates can resolve coarse-grained features on the ice sheet, ATM measurements confirm the implied large bias values.*

Technical Comments
Line 44: doubled Harding et al. (2011) citation
Deleted

Line 68: missing space after the Fair et al. (2024) citation
Fixed

Line 102: "ATM (the Airborne Topographic Mapper) makes laser-altimetry…"
Fixed

Equation 1: I don't think reff is ever explicitly defined in the text
Now defined: *, $r_{eff}$ is the optical effective grain size, corresponding to the radius of a collection of ice spheres that would have the same surface-to-volume ratio as the scattering medium (Grenfell and Warren, 1999),*

Line 246: "… whose normal makes and angle …"
Fixed

Lines 247 and 249: j versus f. I would recommend standardizing the notation
Fixed

Equation 4: the reff notation used up to this point seems to have been replaced with ro
Fixed

Line 329: do the authors mean when the SNR is low? It appears they are pointing to the upper right portion of Figure 3 and the similar error bars between the Pmax=225 (high SNR) and Pmax=90 (low SNR) scenarios.
The referee is correct. Changed to 'low'.

Line 355: here h_li notation is used whereas on line 338 it is hli. The h_li notation appears at other points in the manuscript as well (e.g., Line 388).
The clearer way to say this is h_li (rather than its subscript representation). Fixed.

Line 361: Is "Over" meant to be capitalized?
This is the first word after a full colon, so it should be capitalized.

Line 403: "… values of B0 and rthr that minimize …"
Fixed

Line 418: Scambos et al. (2012) does not appear in the reference list
Reference updated to Haran et al, 2018.

Line 423: "… (Fig. 5c) …"
Fixed

Figures 9 and 10: I thank the authors for homogenizing AVIRIS and AVIRIS-NG in their revised manuscript, but I would suggest also carrying this through the x-axis labels in these two figures.
Fixed

Line 545: "… (Fig. 11a) …"
Fixed

Line 561: "… (Fig. 4) …"

Fixed

Line 565: should there be a unit give to $10^{0.25}$? Perhaps μm?

Fixed, using μm.

Figure 13: In the caption, two different fonts and font sizes are used when referring to Equations on Line 611 and Line 615.

Good eye. Fixed.

Line 632: Missing space prior to the Fair et al. (2024) citation.

Fixed.

Lines 735-736: "… satellite-driven grain-size estimates of providing estimates that would …" could the phrasing here be improved?

Yes, the phrasing can be improved (I hope): "*Another possible data source for corrections of this type would be grain size predictions driven by a grain size-evolution model driven by meteorological data, remote-sensing data, or model output. Unlike grain-size estimates derived purely from satellite measurements, these would not be limited by the availability of cloud-free observations, and might be able to integrate remote-sensing data from multiple sources to reduce the effects of measurement errors.*"

**Referee 2 response, second round**

Thanks much to Dr. Ryan for his second look at our manuscript. We have attended to his comments as described below, with our responses in blue sans-serif font, and all quotes from the revised manuscript in *italics*.

This is an expansive study that describes some innovative ways of correcting ICESat-2 range biases by deriving grain size from several sources of observational data. There are several challenges that the authors had to overcome to complete this study. For example, quirks of individual sensors, conflating processes that influence waveforms, spurious relationships between grain size derived from ATM, AVIRIS, or Sentinel-3 OLCI data. etc. Many sections of the manuscript definitely require close attention when reading but the authors should be commended for their honest and thorough description of their approach. I endorse acceptance of this manuscript in The Cryosphere.

I also appreciate the amount of work that the authors have put into the revised manuscript. I do not have many more comments. Those that remain are mostly associated with the new text that may help improve the style and clarity of the manuscript. Note that my line numbers correspond to the tracked changes version.

Technical comments

L42: Sorry I didn't catch this in my previous review but surely laser altimetry techniques allow efficient measurement of "glacier ice" surface elevations as well? Consider removing "snow-" from this sentence to generalize to both cases.
Fixed.

L58: I think it would be more accurate to say ice-sheet "elevation" changes.
Fixed.

L204: Should be Fair "et al." (2024)
Fixed

L115-120: These sentences would read better if the references were at the end of their respective sentences.
Thanks for the suggestion. I changed the second three sentences to fit this model, but left the first, where the placement of the references is relevant to the meaning of the sentence.

L121-122: Please clarify that this is "airborne" altimeter data since there was a lot of text about ICESat-2 in the previous section which may confuse readers.
Fixed.

L257: Subject-verb agreement issue here. Should be "ATM makes…" if the word "system" is singular.
Fixed.

L552-553: "ablation-zone surfaces" is a little vague since that could include snow or ice depending on the time of year. What about just "observed for glacier ice"?
Changed to "*melting snow and glacier-ice surfaces*"

L535-539: I know it's obvious but probably should define r_eff here as well.
Now defined: *$r_{eff}$ is the optical effective grain size, corresponding to the radius of a collection of ice spheres that would have the same surface-to-volume ratio as the scattering medium (Grenfell and Warren, 1999)*,

L658: "cases"
Fixed.

L853: Explain why infrared light does not penetrate as far as green light
We added: *"because of the stronger attenuation of infrared light by ice (Warren 2008)*, "

L1210-1216: It would be useful to briefly explain the dashed red line in the caption.
Fixed.

L1391: "are" biased?
Fixed.

L1744: But didn't you use the 1 km OLCI products?
Fixed.

L1745: Check whether Landsat should be all capitalized
Fixed.

---

## Author Response (AR3)

Dear Dr. Sørensen,

Thanks for checking over the manuscript again, and for all your help.  I've made the following changes, as requested:

L. 17: form -> from.

> Fixed

L. 44: Harding reference still appears twice?

> Fixed

L. 158: space missing before the ~

> Try as I might, I could not find a ~ that did not have a space in front of it.  This may be a problem with how the manuscript is rendered on either your or my computer, but we'll get this sorted out in the proof stage.

L. 263: space missing before reference

> Fixed

L 648 and 649: bias -> biases ?

> Fixed

L. 651: "spread of biases of the ICESat-2 biases" -> "spread of the ICESat-2 biases" ?

> Fixed.

All the best, and happy New Year.

--Ben